# TPO: ALIGNING LARGE LANGUAGE MODELS WITH MULTI-BRANCH & MULTI-STEP PREFERENCE TREES

**Weibin Liao♠, Xu Chu♠◇♡\*, Yasha Wang♠♡**

♠School of Computer Science, Peking University
◇Center on Frontiers of Computing Studies, Peking University
♡National Research and Engineering Center of Software Engineering, Peking University
✉ liaoweibin@stu.pku.edu.cn, chu_xu@pku.edu.cn
⬡ https://github.com/MrBlankness/TPO.git

## ABSTRACT

In the domain of complex reasoning tasks, such as mathematical reasoning, recent advancements have proposed the use of Direct Preference Optimization (DPO) to suppress output of dispreferred responses, thereby enhancing the long-chain reasoning capabilities of large language models (LLMs). To this end, these studies employed LLMs to generate preference trees via Tree-of-thoughts (ToT) and sample the paired preference responses required by the DPO algorithm. However, the DPO algorithm based on binary preference optimization was unable to learn multiple responses with varying degrees of preference/dispreference that provided by the preference trees, resulting in incomplete preference learning. In this work, we introduce **T**ree **P**reference **O**ptimization (TPO), which does not sample paired preference responses from the preference tree; instead, it directly learns from the entire preference tree during the fine-tuning. Specifically, TPO formulates the language model alignment as a *Preference List Ranking* problem, where the policy can potentially learn more effectively from a ranked preference list of responses given the prompt. In addition, to further assist LLMs in identifying discriminative steps within long-chain reasoning and increase the relative reward margin in the preference list, TPO utilizes *Adaptive Step Reward* to adjust the reward values of each step in the trajectory for performing fine-grained preference optimization. We carry out extensive experiments on mathematical reasoning tasks to evaluate TPO. The experimental results indicate that TPO consistently outperforms DPO across five publicly large language models on four datasets.

## 1 INTRODUCTION

Long-chain reasoning task (Wei et al., 2022; Xiong et al., 2024), such as commonsense reasoning and math reasoning, is one of the critical capabilities in large language models (LLMs) (Lai et al., 2024). This task is particularly challenging as it often involves numerous reasoning steps. Any mistake in these steps can lead to an incorrect final answer. Initially, some studies utilized various data augmentation techniques during the supervised fine-tuning (SFT) phase to enhance the reasoning capabilities of LLMs (Shao et al., 2024; Tang et al., 2024; Xin et al., 2024). However, a phenomenon of pessimism suggests that the positive feedback provided by SFT alone cannot prevent LLMs from generating erroneous reasoning pathways. Hong et al. (2024) indicated that, during the SFT phase, as the probability of preferred outputs increases, the probability of dispreferred outputs also rises. This phenomenon makes the models more prone to errors in long-chain reasoning. Consequently, it is necessary to develop methods to mitigate the likelihood of dispreferred outputs.

Recently, Direct Preference Optimization (DPO) (Rafailov et al., 2023) has been proposed for aligning LLMs using paired preference data. Compared to the traditional Reinforcement Learning from Human Feedback (RLHF) (Christiano et al., 2017) framework, DPO has gained popularity due to its simplicity and reduced memory requirements. Recent studies have utilized DPO to suppress

---

\*Corresponding Author.

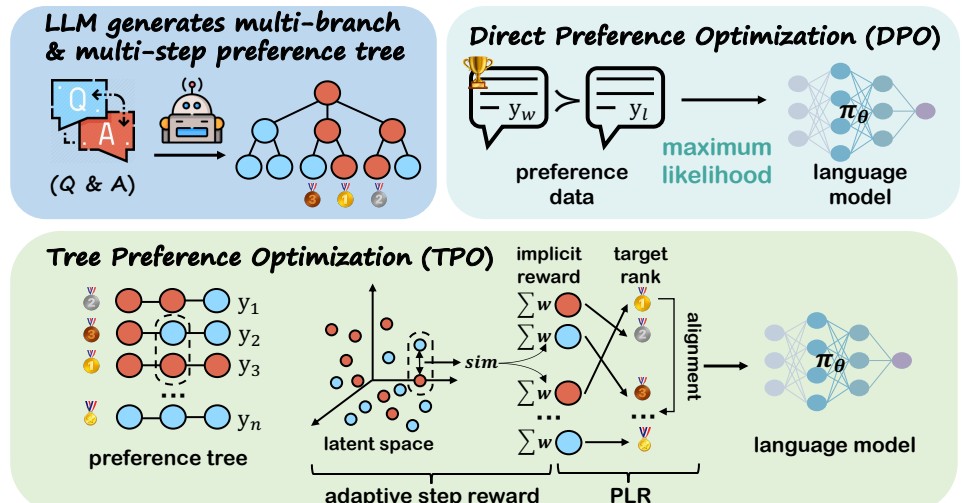

Figure 1: The framework of `TPO`: TPO regards preference modeling as a more general *Preference List Ranking* (PLR) problem and employs an *Adaptive Step Reward* for achieving finer-grained preference optimization.

dispreferred responses in LLM outputs (Lai et al., 2024; Xie et al., 2024). For this purpose, they have employed LLMs to generate preference trees via Tree-of-thoughts (ToT) (Yao et al., 2023), and based on the inherent characteristics of the preference trees, they have collected the paired preference data required for DPO training. In general, existing works employ heuristic methods to sample paired preference data, typically manifesting as the **random selection** of reasoning trajectories that can/cannot correctly answer the question as preferred/dispreferred responses (Jiao et al., 2024). Alternatively, reasoning trajectories may be **manually selected** (particularly evident in the choice of dispreferred responses) to ensure the quality of the data (Lai et al., 2024); however, this approach further introduces costly manual annotation efforts.

Although sampling-based strategies have been proven effective, we consider them to be a **inferior solution** that is constrained by the fact that DPO supports only binary preference data input. We still seek a preference learning algorithm tailored specifically for preference trees. This is demonstrated by the following points:

1. Preference trees typically yield unbalanced preference responses, with a large number of dispreferred responses being randomly filtered out and not incorporated into the DPO, resulting in a **low data utilization efficiency** for the model. Additionally, due to the inherent nature of the tree structure, these dispreferred responses encompass varying degrees of reward. For instance, although neither response $y_1$ nor $y_2$ leads to the correct outcome, $y_2$ may contain more correct reasoning steps than $y_1$, resulting in an inequality among dispreferred responses. We contend that *DPO based on binary rewards is unable to explore the critical information within failure trajectories, we propose to introduce preferences with varying reward values to facilitate more robust preference optimization.*

2. The responses in the preference tree may share a portion of sub-trajectories, which leads to a **lower reward margin** between preferences, especially when a large number of shared sub-trajectories are present. This issue has not been considered in the existing DPO algorithm. We contend that *the lower reward margin may prevent the model from discerning the differences between preference pairs. Consequently, we need to adaptively adjust the step rewards to enable fine-grained optimization.*

Motivated by the aforementioned points, in this work, we introduce **T**ree **P**reference **O**ptimization (`TPO`), which does not sample paired preference responses from the preference tree; instead, it directly learns from the entire preference tree. Specifically, `TPO` decouples the preference tree into multi-branch & multi-step responses and performs preference optimization. To align LLMs from

multi-branch preferences, TPO formulates LM alignment as a more general *Preference List Ranking* problem and establishes a connection between LM alignment and Learn-to-Rank (Liu et al., 2009; Li et al., 2024b), enabling the LLMs to learn alignment more effectively from preference lists. To align LLMs from multi-step preferences, TPO proposes the *Adaptive Step Reward* that adjusts the reward value of each step based on the correlation scores between pairs of steps. This mechanism aids LLMs in conducting preference optimization from more discriminative steps. Although TPO is a preference learning algorithm that introduces the reward value, unlike traditional reinforcement learning algorithms, the reward used in TPO is not obtained through a learning-based algorithm, implying a lower learning variance.

Our contributions are summarized as follows:

1. In the context of long-chain reasoning tasks, we consider issues related to current DPO algorithms, specifically its low data utilization efficiency when aligning large language models on preference trees, as well as the inability of DPO based on binary rewards to explore critical information in failure trajectories. To tackle these challenges, we propose TPO, the first preference optimization algorithm designed specifically for tree-structured preference data.

2. We propose the *Preference List Ranking* optimization objective that connects LM alignment and Learn-to-Rank, establishing a framework that enables preference modeling from responses with varying reward values. In addition, we propose the *Adaptive Step Reward* mechanism, which addresses the issue of reduced reward margin between responses generated from preference trees due to shared sub-trajectories.

3. We conduct extensive experiments to validate the effectiveness of TPO and discuss its generalization to out-of-distribution datasets.

## 2 PRELIMINARIES

### 2.1 DIRECT PREFERENCE OPTIMIZATION

Reinforcement Learning from Human Feedback (RLHF) (Christiano et al., 2017) is an effective method for enhancing the robustness, authenticity, and safety of LLMs (Ouyang et al., 2022), which directly optimizes LLMs according to human preferences by maximizing the reward value of the model's responses. The reward function is defined based on the Bradley-Terry (BT) model (Bradley & Terry, 1952) of preferences. Specifically, for preferred response $y_w$ and dispreferred response $y_l$ under the same prompt $x$ and data distribution $\mathcal{D}$, the BT model stipulates that the human preference distribution $p^*$ can be expressed as:

$$p^*_{\mathcal{D}}(y_w \succ y_l \mid x) = \sigma\left(r^*(x, y_w) - r^*(x, y_l)\right) \qquad (1)$$

where $p^*_{\mathcal{D}}(y_w \succ y_l)$ denotes the probability that $y_w$ is preferred against $y_l$, $\sigma(x) = \frac{1}{1+\exp(-x)}$ denotes the sigmoid function, and $r^*$ denotes some latent reward model, which we do not have access to. The alignment of language models is commonly regarded as an optimization problem with a Kullback-Leibler (KL) constraint on reward values, formalized as follows:

$$max\mathbb{E}_{x \sim \mathcal{D}, y \sim \pi_\theta(y|x)}\left[r^*(x, y)\right]$$
$$s.t.\mathbb{E}_{x \sim \mathcal{D}}\mathbb{D}_{KL}\left[\pi_\theta(y \mid x) \| \pi_{ref}(y \mid x)\right] \leq \sigma \qquad (2)$$

where $\pi_\theta$ denotes the aligned policy model, $\pi_{ref}$ denotes the reference policy model. To prevent reward hacking and ensure that $\pi_\theta$ does not deviate too much from the $\pi_{ref}$ (Amodei et al., 2016), a regularization term is typically added to the objective function (Stiennon et al., 2020), the problem is transformed into:

$$max\mathbb{E}_{x \sim \mathcal{D}, y \sim \pi(y|x)}\left[r^*(x, y)\right] - \beta\mathbb{D}_{KL}\left[\pi_\theta(y \mid x) \| \pi_{ref}(y \mid x)\right] \qquad (3)$$

where the hyperparameter $\beta$ controls the KL divergence between $\pi_\theta$ and $\pi_{ref}$. In general, RLHF encompasses two training phases, including reward model training, and policy model training. However, the ultimate performance of RLHF is highly sensitive to various hyperparameters across these

two phases, requiring careful tuning. To circumvent this complex training process, Rafailov et al. (2023) introduced Direct Preference Optimization (DPO), which directly utilizes paired preference data to optimize the policy model, bypassing the reward modeling stage by directly substituting this closed-form solution in Eq. 1. Specifically, given an input prompt $x$ and a pair of preference data $(y_w, y_l)$, the goal of DPO is to maximize the probability of the preferred response $y_w$ and minimize the probability of the dispreferred response $y_l$, yielding the following DPO objective:

$$\mathcal{L}_{\text{DPO}}(\pi_\theta; \pi_{\text{ref}}) = -\mathbb{E}_{(x, y_w, y_l) \sim \mathcal{D}} \left[ \log \sigma \left( \beta \log \frac{\pi_\theta(y_w \mid x)}{\pi_{\text{ref}}(y_w \mid x)} - \beta \log \frac{\pi_\theta(y_l \mid x)}{\pi_{\text{ref}}(y_l \mid x)} \right) \right] \quad (4)$$

where the $\beta \log \frac{\pi_\theta(y|x)}{\pi_{\text{ref}}(y|x)}$ can be regarded as an "implicit reward" (Rafailov et al., 2023), and the objective of DPO is to align the "implicit reward" directly with human preference data.

## 2.2 TREE-STRUCTURED REASONING POLICY FOR LLM

**Multi-step Reasoning**   Following the standard reasoning setup of LLMs, given a policy $\pi$ instantiated by LLM and an input prompt $x$, $\pi$ can step-by-step generate a trajectory of reasoning steps $y = (s_1, \cdots, s_K) \sim \pi(\cdot|x)$ by autoregressively predicting the next token. The standard reasoning setup assumes that $y$ encompasses the complete list of reasoning steps, with each step $s_k$ comprising multiple tokens. The step-by-step long-chain reasoning process is most famously used in Chain-of-Thought (CoT) (Wei et al., 2022).

**Multi-branch Reasoning**   Self-Consistency (Wang et al., 2023) was the first to introduce multi-branch reasoning. Given an input prompt $x$, the policy $\pi$ generates $N$ trajectories of reasoning steps $\mathbf{y} = (y_1, \cdots, y_N) \sim \pi(\cdot|x)$, where $y_i = (s_1^i, \cdots, s_K^i)$. Ultimately, Self-Consistency selects the most probable final answer by marginalizing over the reasoning trajectories.

**Tree: Multi-branch & Multi-step Reasoning**   Recent works have further extended CoT and Self-Consistency to a tree-like structure, referred to as the Tree-of-Thoughts (ToT) (Yao et al., 2023). Specifically, ToT no longer confines its application to the initial prompt but extends to engaging in branching reasoning at any intermediate state subsequent to given steps. Given the state $\mathbf{S} = [x, s_{1, \cdots, k-1}]$ of an LLM in the reasoning trajectory, ToT employs a Thought Generator $G(\pi, \mathbf{S}, N)$ to propose $N$ next planning steps $[s_k^{(1)}, \cdots, s_k^{(N)}]$. Compared to CoT, ToT possesses a broader space for cognitive exploration and can circumvent the generation of repetitive responses within the same context.

## 3 METHODOLOGY

We propose **T**ree **P**reference **O**ptimization (`TPO`), a preference learning algorithm tailored for preference trees generated by LLMs via Tree-of-Thoughts. `TPO` learns a preference list with varying reward values using a *Preference List Ranking* objective, and utilizes *Adaptive Step Reward* to achieve fine-grained alignment of step rewards.

## 3.1 ALIGNING LLMs FROM MULTI-BRANCH PREFERENCES

**Problem Definition**   `TPO` defines the dataset $\mathcal{D} = (x^{(i)}, \mathbf{y}^{(i)}, \mathbf{v}^{(i)})_{i=1}^M$ with $M$ samples: given a prompt $x$, there is a response list $\mathbf{y} = (y_1, \cdots, y_N)$ of size $N$, and each response $y$ is associated with a reward value $v$. The responses $\mathbf{y}$ are generated by the policy $\pi$, while the reward values $\mathbf{v}$ are derived from human raters or an inaccessible reward model. Typically, $\mathbf{v} = (v_1, \cdots, v_N) \in [0, 1]^N$, with higher reward values indicating better responses.

**Connection Between LM Alignment and Learn-to-Rank**   `TPO` follows the definition of the classic Learning-to-Rank (LTR) (Liu et al., 2009; Li et al., 2023; Liao et al., 2024c) problem: the optimization objective is to learn a ranking model that outputs the relevance *scores* for all *documents* given a *query*. In the context of LM alignment, `TPO` treats prompt $x$ as the *query* and responses $\mathbf{y}$ as *documents*. Inspired by Rafailov et al. (2023), `TPO` further regards the normalized "implicit reward"

(denoted as $\mathbf{r} = (r_1, \cdots, r_N) = (\beta \log \frac{\pi_\theta(y_1|x)}{\pi_{\text{ref}}(y_1|x)}, \cdots, \beta \log \frac{\pi_\theta(y_N|x)}{\pi_{\text{ref}}(y_N|x)}) \in [0,1]^N$) as an evaluation of the model's relevance *scores* for $x$ and $\mathbf{y}$. Overall, TPO establishes the following connection between LM alignment and Learn-to-Rank.

$$
\begin{aligned}
&\text{LM Alignment} \leftarrow \text{Learn-to-Rank} \\
&\texttt{prompt}, x := \texttt{query} \\
&\texttt{responses}, \mathbf{y} := \texttt{documents} \\
&\texttt{implicit rewards}, \mathbf{r} := \texttt{scores}
\end{aligned}
\tag{5}
$$

**Preference List Ranking** The LTR algorithm defines the ranking loss function based on rewards $\mathbf{v}$ of responses $\mathbf{y}$ and predicted scores $\mathbf{r}$, to train the model $\pi$:

$$
\mathcal{L}_{LTR} = \mathbb{E}_{(x,\mathbf{y},\mathbf{v})\sim\mathcal{D}} [l(\mathbf{v}, \mathbf{r})].
\tag{6}
$$

where $l$ is the loss function. TPO proposes the *Preference Ranking Loss* $\mathcal{L}_{PLR}$ to instantiate $l$. Specifically, TPO utilizes the relative reward margin between each pair of preferences in the preference list to train the policy $\pi$. To further consider the absolute position of preferences within the list, inspired by Burges et al. (2006), TPO introduces the Lambda Weight (Burges et al., 2006) to optimize $\mathcal{L}_{PRL}$, in order to perceive the impact brought about by the change in the positions of two preferences. Ultimately, $\mathcal{L}_{PRL}$ is mathematically represented as follows:

$$
\mathcal{L}_{PRL} = -\mathbb{E}_{x,\mathbf{y},\mathbf{v}\sim\mathcal{D}} \left[ \lambda_{i,j} \sum_{v_i > v_j} \log \sigma(r_i - r_j) \right]
\tag{7}
$$

$$
\lambda_{i,j} = |2^{v_i} - 2^{v_j}| \cdot \left| \frac{1}{log(1+\tau(i))} - \frac{1}{log(1+\tau(j))} \right|
\tag{8}
$$

where $\tau(i)$ is the ranking position of $y_i$ in the ranking permutation induced by $\mathbf{r}$. For more detailed information on Lambda Weight, please refer to Burges et al. (2006). It is worth noting that when the length of the preference list $N = 2$ and the Lambda Weight is not introduced, $\mathcal{L}_{PRL}$ degenerates into the naive DPO loss.

## 3.2 ALIGNING LLMs FROM MULTI-STEP PREFERENCES

**Problem Definition** TPO follows the definition of multi-step reasoning as described in Sec. 2.2, introducing $y = (s_1, s_2, \cdots, s_K)$ consisting of $K$ steps. Due to the characteristics inherent in tree-structured reasoning, for any two reasoning trajectories $y_i = (s_1^i, \cdots, s_K^i)$ and $y_j = (s_1^j, \cdots, s_K^j)$, (To simplify, TPO assumes that $y_i$ and $y_j$ possess steps of equal length.) there exist sub-trajectories which are *content sharing* or *action sharing*.

- *Steps of content sharing*: $y_i$ and $y_j$ have traversed the same sub-trajectory $(s_1, \cdots, s_{k-1})$ and branched off at state $\mathbf{S}_k$. Due to the $(s_k^i \neq s_k^j) \sim \pi(\cdot|x)$, which resulting in $\mathbf{S}_k^i \neq \mathbf{S}_k^j$.

- *Steps of action sharing*: Expanding on *content sharing*, even though the $(s_k^i \neq s_k^j)$, the high degree of semantic similarity or the execution of identical actions results in $\mathbf{S}_k^i = \mathbf{S}_k^j$.

**Adaptive Step Reward** In the naive DPO algorithm, the "implicit reward" margin is step-independent, that is, for responses $y_i$ and $y_j$, the "implicit reward" margin is mathematically defined as follows:

$$
\mathcal{RM} = \beta \log \frac{\pi_\theta(y_i \mid x)}{\pi_{\text{ref}}(y_i \mid x)} - \beta \log \frac{\pi_\theta(y_j \mid x)}{\pi_{\text{ref}}(y_j \mid x)} = \sum_{k=1}^{K} (\beta \log \frac{\pi_\theta(s_k^i \mid x)}{\pi_{\text{ref}}(s_k^i \mid x)} - \beta \log \frac{\pi_\theta(s_k^j \mid x)}{\pi_{\text{ref}}(s_k^j \mid x)})
\tag{9}
$$

To mitigate the reduced reward margin resulting from shared steps, TPO introduces the *Adaptive Step Reward* mechanism to discriminatively assign rewards for each step. Specifically, TPO employs adaptive weight $w$ to adjust reward margin between step pairs, and instantiates $w$ as cosine similarity in the semantic space. The adaptive $\mathcal{RM}$ can be mathematically expressed as follows:

$$\mathcal{RM} = \sum_{k=1}^{K}((1 + \frac{emb(s_k^i) \cdot emb(s_k^j)}{\|emb(s_k^i)\|\|emb(s_k^j)\|}) \cdot (\beta \log \frac{\pi_\theta(s_k^i \mid x)}{\pi_{\text{ref}}(s_k^i \mid x)} - \beta \log \frac{\pi_\theta(s_k^j \mid x)}{\pi_{\text{ref}}(s_k^j \mid x)})) \qquad (10)$$

where $emb(\cdot)$ is the operation for semantic vectors generation. It is worth noting that when $s_k^i$ and $s_k^j$ manifest as steps of content sharing, the current step pairs yields a provided $\mathcal{RM} = 0$ due to the $\beta \log \frac{\pi_\theta(s_k^i|x)}{\pi_{\text{ref}}(s_k^i|x)} - \beta \log \frac{\pi_\theta(s_k^j|x)}{\pi_{\text{ref}}(s_k^j|x)}) = 0$.

Ultimately, the overall algorithm of `TPO` is detailed in Appendix Alg. 1.

## 4 EXPERIMENTS

### 4.1 EXPERIMENTAL SETUPS

**Network Architecture** Our experiments were based on various base models, including Qwen2 models (Bai et al., 2023) of various sizes (Qwen2-1.5B-Instruct and Qwen2-7B-Instruct), Meta-Llama-3-8B-Instruct (Touvron et al., 2023), Mistral-7B-Instruct-v0.3 (Jiang et al., 2023) and the DeepSeekMath-7B-Instruct (Shao et al., 2024) that has been specifically fine-tuned for mathematical tasks. We also introduced DeepSeekMath-7B-RL (Shao et al., 2024), which underwent reinforcement learning by Shao et al. (2024), as the baseline model.

**Training Datasets** Typically, when faced with complex mathematical problems, LLMs struggle to arrive at the correct final answer even when employing ToT methods. To ensure that the preference tree can generate trajectories capable of reasoning to the correct answer, we have expanded upon the existing dataset. Lai et al. (2024) proposed a dataset that provides 10,795 paired preference data, completely composed of mathematical problems, with complete correct and incorrect reasoning trajectories provided for each problem. As shown in Fig. 2(a), starting from any intermediate step in the correct reasoning trajectory, we utilized Qwen2-7B-Instrcut (Bai et al., 2023) for further step-by-step reasoning resulting in trajectories with varying degrees of preference. Ultimately, we collected 10 trajectories for each problem, including at least one correct trajectory and one incorrect trajectory from original data (Lai et al., 2024), with the remaining eight were generated by Qwen2-7B-Instrcut. We utilized ChatGPT to assign scores ranging from -100 to 100 for each trajectory in order to obtain the rewards for these trajectories. To avoid incorrect judgments by ChatGPT, we provided the correct trajectory as a reference and employed ReACT as shown in Fig. 2(b). The prompts for data generation and ChatGPT scoring can be found in the Appendix. B. Fig. 2(c) illustrates the distribution of reward values for this dataset. The statistical findings reveal that we ultimately gathered preference data corresponding to a reward value distribution of $53.74 \pm 69.27$. It is noteworthy that Bai et al. (2023) shows that the dataset was derived from the MetaMath (Yu et al., 2023), MMIQC (Liu & Yao, 2024), and AQuA (Ling et al., 2017) datasets. We have ensured that these datasets do not overlap with the subsequent evaluation data.

**Evaluation Datasets** We introduced three types of tasks, **Math** (in-distribution), **Coding** and **Reasoning** (out-of-distribution), to assess the effectiveness of `TPO`. For the **Math** tasks, we considered the following datasets: MATH (Hendrycks et al., 2021), SVAMP (Patel et al., 2021), AS-Div (Miao et al., 2021) and GSM-Plus (Li et al., 2024a). For the **Coding** tasks, we considered the HumanEval (Chen et al., 2021) and MBPP (Austin et al., 2021) datasets. For **Reasoning** task, we considered the BBH (Suzgun et al., 2023) and MMLU (Hendrycks et al.) datasets. It is worth noting that BBH tasks require multi-step reasoning, whereas MMLU does not. The prompts used for evaluating these datasets can be found in the Appendix. B. We evaluated these datasets with pass@1 accuracy.

**Implement Details** We performed the `TPO` and DPO on the models mentioned above. We used the PyTorch library to implement all the algorithms based on the open-source HuggingFace transformers (Wolf, 2019) and Transformer Reinforcement Learning (TRL) (von Werra et al., 2020). The experiments were conducted on 8 NVIDIA-RTX3090-24GB GPUs. For each experimental setup, we trained the model for 1 epoch, using a batch size of 1 for each GPU. The learning rate was set to

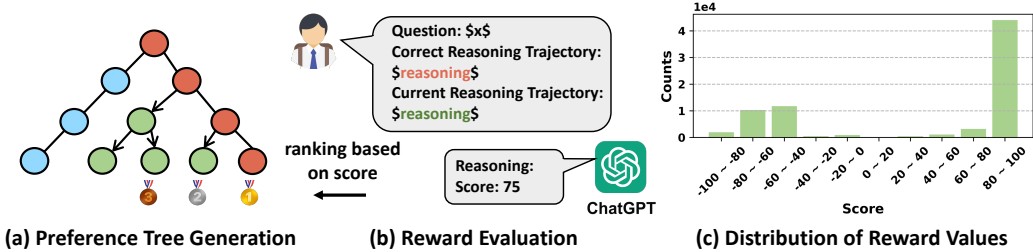

Figure 2: (a) illustrates the data generation pipeline we used, where we start from the intermediate steps of the original correct reasoning trajectories and generate new reasoning trajectories step by step. The ● steps represent preferred reasoning steps, the ● steps denote dispreferred reasoning steps, and the ● steps indicate reasoning steps with unknown preference. (b) shows how we introduced ChatGPT to score each reasoning trajectory, with scores ranging from $\in [-100, 100]$. We provided ChatGPT with correct reasoning trajectories as a reference and employed ReACT to improve score credibility. (c) presents the distribution of reasoning trajectories across various score intervals.

5e-7. The hyperparameter $\beta$ used in Eq. 4 for DPO was set to 0.5. We utilized the AdamW optimizer and a cosine learning rate scheduler, with a warm-up ratio set to 0.1.

## 4.2 EXPERIMENT RESULTS

**Results on Math Task** We conducted evaluations on four mathematics reasoning datasets to verify the performance of TPO on in-distribution datasets. We employed the CoT (Wei et al., 2022) strategy for reasoning without using any demonstrations. Results are shown in Table. 1. We summarize the key takeaways as follows:

TPO comprehensively outperformed the SFT and DPO algorithm across all datasets, across various LLM size settings (Qwen2-1.5B-Instruct and Qwen2-7B-Instruct) and whether the LLMs were fine-tuned in the domain of mathematics (Qwen2-7B-Instruct and DeepSeekMath-7B-Instruct). Similar experimental results were also observed on Meta-Llama-3-8B-Instruct and Mistral-7B-Instruct-v0.3 (see Appendix Table. 4). In many cases, TPO also surpassed existing specialized reinforcement learning methods (DeepSeekMath-7B-RL *vs.* DeepSeekMath-7B-Instruct+TPO), with only a slight disadvantage on the ASDiv dataset. We further discovered that TPO can help LLMs exceed baseline models that are 5× larger. For instance, on the SVAMP dataset, the performance of Qwen2-1.5B-Instruct was significantly inferior to that of Qwen2-7B-Instruct. However, after fine-tuning with TPO, Qwen2-1.5B-Instruct+TPO outperformed Qwen2-7B-Instruct by 1.7% in accuracy. DeepSeekMath-7B-Instruct, fine-tuned with TPO, has significantly surpassed GPT-3.5 Turbo.

**Results on Coding and Reasoning Task** We further conducted evaluations on two coding datasets and two reasoning datasets to verify the performance of TPO on out-of-distribution datasets. Results are shown in Table. 2. We summarize the key takeaways as follows:

For coding tasks, in the vast majority of cases, TPO aided in improving the performance of LLMs on out-of-distribution datasets, surpassing the DPO algorithm. However, on the HumanEval dataset, Qwen2-1.5B-Instruct and Qwen2-7B-Instruct exhibited a decline in performance after undergoing TPO, a phenomenon similarly observed with the DPO approach. Notably, DeepSeekMath-7B-Instruct experienced an improvement in performance on the HumanEval dataset after the preference alignment, regardless of whether the DPO or TPO algorithm was used. We **speculate** that the cause of this phenomenon is that the Qwen2 series models have already been fine-tuned on the HumanEval dataset, leading to "catastrophic forgetting" (Xuhong et al., 2018; Liao et al., 2022; 2024b) in the Qwen2 models after the preference alignment, where they forgot the coding knowledge originally learned on the HumanEval dataset. In contrast, DeepSeekMath-7B-Instruct is a model specifically designed for mathematical reasoning and did not acquire coding knowledge during its previous training phases.

For reasoning tasks, we observe that TPO once again surpasses existing algorithms on the BBH dataset. However, on the MMLU dataset, the performance across all algorithms is similar. We

Table 1: Experimental results on **Math** task (In-Distribution) using the SFT, DPO and `TPO` algorithm. The best results for each large language model setting are indicated in **bold**. The best results across all settings are highlighted with a background, while the second-best results are indicated with a background. We report results using pass@1 accuracy.

| LLMs | size | open | general | MATH | SVAMP | ASDiv | GSM-Plus | Avg. |
|---|---|---|---|---|---|---|---|---|
| Qwen2-1.5B-Instruct | 1.5B | ✓ | ✓ | 19.52 | 23.90 | 35.76 | 20.05 | 24.81 |
| Qwen2-1.5B-Instruct+SFT | 1.5B | ✓ | ✓ | 20.80 | 28.77 | 38.32 | 21.87 | 27.44 |
| Qwen2-1.5B-Instruct+DPO | 1.5B | ✓ | ✓ | 20.98 | 29.30 | 40.13 | 21.52 | 27.98 |
| Qwen2-1.5B-Instruct+TPO | 1.5B | ✓ | ✓ | **22.88** | **35.60** | **46.28** | **24.12** | **32.22** |
| Qwen2-7B-Instruct | 7B | ✓ | ✓ | 53.92 | 33.90 | 48.38 | 44.72 | 45.23 |
| Qwen2-7B-Instruct+SFT | 7B | ✓ | ✓ | 54.92 | 46.40 | 53.28 | 45.40 | 50.00 |
| Qwen2-7B-Instruct+DPO | 7B | ✓ | ✓ | 54.26 | 44.69 | 54.32 | 50.28 | 50.89 |
| Qwen2-7B-Instruct+TPO | 7B | ✓ | ✓ | 55.46 | **48.20** | **59.22** | **54.82** | **54.43** |
| DeepSeekMath-7B-Instruct | 7B | ✓ | ✗ | 43.92 | 84.10 | 90.10 | 59.78 | 69.48 |
| DeepSeekMath-7B-RL | 7B | ✓ | ✗ | 50.70 | 86.70 | 90.94 | 64.07 | 73.10 |
| DeepSeekMath-7B-Instruct+SFT | 7B | ✓ | ✗ | 45.18 | 84.30 | 90.45 | 61.09 | 70.26 |
| DeepSeekMath-7B-Instruct+DPO | 7B | ✓ | ✗ | 48.66 | 85.98 | 90.16 | 62.86 | 71.92 |
| DeepSeekMath-7B-Instruct+TPO | 7B | ✓ | ✗ | **51.30** | **86.80** | 90.61 | **64.73** | **73.36** |
| GPT-3.5 Turbo | - | ✗ | ✓ | 37.80 | 83.00 | 90.60 | 61.20 | 68.15 |
| GPT-4 | - | ✗ | ✓ | **69.70** | **94.80** | **92.60** | **85.60** | **85.68** |

Table 2: Experimental results on **Coding** and **Reasoning** task (Out-of-Distribution) using the SFT, DPO and `TPO` algorithm. The best results for each large language model setting are indicated in **bold**. We report results using pass@1 accuracy.

| LLMs | size | open | general | Coding | | Reasoning | |
|---|---|---|---|---|---|---|---|
| | | | | MBPP | HumanEval | BBH | MMLU |
| Qwen2-1.5B-Instruct | 1.5B | ✓ | ✓ | 45.11 | **46.34** | 32.76 | 55.97 |
| Qwen2-1.5B-Instruct+SFT | 1.5B | ✓ | ✓ | 45.63 | 46.20 | 33.10 | 55.90 |
| Qwen2-1.5B-Instruct+DPO | 1.5B | ✓ | ✓ | 45.11 | 44.51 | 34.72 | **56.03** |
| Qwen2-1.5B-Instruct+TPO | 1.5B | ✓ | ✓ | **46.62** | 43.90 | **37.49** | 55.99 |
| Qwen2-7B-Instruct | 7B | ✓ | ✓ | 58.90 | **75.00** | 62.43 | **70.78** |
| Qwen2-7B-Instruct+SFT | 7B | ✓ | ✓ | 62.91 | 77.44 | 64.75 | 70.74 |
| Qwen2-7B-Instruct+DPO | 7B | ✓ | ✓ | 59.11 | 68.32 | 65.58 | 70.58 |
| Qwen2-7B-Instruct+TPO | 7B | ✓ | ✓ | **61.65** | 65.85 | **69.62** | 70.63 |
| DeepSeekMath-7B-Instruct | 7B | ✓ | ✓ | 60.90 | 56.10 | 61.85 | 54.44 |
| DeepSeekMath-7B-Instruct+SFT | 7B | ✓ | ✓ | 59.65 | 56.10 | 62.79 | 54.56 |
| DeepSeekMath-7B-RL | 7B | ✓ | ✓ | 65.91 | 56.10 | 62.74 | 54.98 |
| DeepSeekMath-7B-Instruct+DPO | 7B | ✓ | ✓ | 63.26 | 57.25 | 62.68 | 54.22 |
| DeepSeekMath-7B-Instruct+TPO | 7B | ✓ | ✓ | **66.42** | **59.15** | **62.99** | **54.99** |
| GPT-3.5 Turbo | - | ✗ | ✓ | 82.50 | 76.80 | 70.10 | 70.00 |
| GPT-4 | - | ✗ | ✓ | **83.50** | **85.40** | **86.70** | **86.40** |

attribute this to `TPO` acquiring long-chain reasoning abilities during fine-tuning on mathematical tasks, which generalize effectively to other domains such as coding and reasoning. However, since MMLU tasks do not require long-chain reasoning, the `TPO` algorithm does not achieve performance improvements on this dataset. Notably, `TPO` does not lead to performance degradation of LLMs on MMLU tasks, indicating that commonsense knowledge is preserved during the `TPO` fine-tuning process.

Table 3: Ablation Studies of TPO on **Math** tasks. The green font indicates the performance loss incurred after the removal of the respective module. Results show that the absence of any module leads to a degradation in performance.

| Methods | MATH | SVAMP | ASDiv | GSM-Plus | Avg. |
|---|---|---|---|---|---|
| Qwen2-7B-Instruct | | | | | |
| TPO | 55.46 | 48.20 | 59.22 | 54.82 | 54.43 |
|   w/o *Adaptive Step Reward* | 55.08(-0.38) | 47.84(-0.36) | 58.72(-0.50) | 54.36(-0.46) | 54.00(-0.43) |
|   w/o *Preference List Ranking* | 54.36(-1.10) | 45.67(-2.53) | 56.84(-2.38) | 51.10(-3.72) | 51.99(-2.44) |
| DeepSeekMath-7B-Instruct | | | | | |
| TPO | 51.30 | 86.80 | 90.61 | 64.73 | 73.36 |
|   w/o *Adaptive Step Reward* | 50.92(-0.38) | 86.68(-0.12) | 90.53(-0.08) | 64.32(-0.41) | 73.11(-0.25) |
|   w/o *Preference List Ranking* | 49.11(-2.19) | 86.29(-0.51) | 90.20(-0.41) | 63.02(-1.71) | 72.16(-1.20) |

**Ablation Studies**   We verified the effectiveness of each module by removing some modules from TPO and evaluated the modified models using the Qwen2-7B-Instruct and DeepSeekMath-7B-Instruct across four mathematical reasoning datasets. The experimental results were presented in Table. 3. The results indicated that the absence of both the *Adaptive Step Reward* and the *Preference List Ranking* modules leads to a degradation in performance of TPO, with the *Preference List Ranking* module's removal resulting in an average performance of 2.44%↓ and 1.40%↓. These results suggest that the *Preference List Ranking* module aids LLMs in learning from a wider variety of preference lists with different reward values, thereby facilitating more robust preference alignment. Regarding the *Adaptive Step Reward*, we provide further discussion through t-test and case studies in the Appendix. C.4 and Appendix. C.5.

## 4.3 ANALYSIS OF DISPREFERRED RESPONSES

**Comparison of DPO with Varying Reward Values**   We performed DPO using preference pairs with different reward values and evaluated the results using Qwen2-7B-Instruct on the ASDiv and GSM-Plus datasets. Specifically, we employed correct reasoning trajectories as preferred responses and sampled dispreferred responses with different reward distributions sampled from incorrect trajectories. The mean reward values with corresponding standard deviations were $[7.4\pm67.7, 56.3\pm50.9, 75.4\pm35.5, 86.4\pm19.5]$. Our experimental results are presented in Fig. 3(a). The results indicate that dispreferred responses with different reward values have varying degrees of impact on the model's performance. Fig. 3(a) shows that when dispreferred responses with lower mean rewards (strong dispreference) or higher mean rewards (weak dispreference) are used, the performance of DPO is inferior. However, the best performance of DPO is observed when dispreferred responses with a moderate mean reward are used. We argue that some dispreferred responses with lower rewards are less valuable for learning due to their significant discrepancy from preferred responses. Conversely, dispreferred responses with higher rewards pose challenges for the DPO algorithm to learn because of their smaller difference from preferred responses. Therefore, it is necessary to select dispreferred responses with moderate rewards to facilitate more effective DPO learning. Nonetheless, TPO still outperforms all DPO baselines, suggesting that introducing more dispreferred responses and aligning language models from preference lists with different reward values concurrently contributes to stronger preference learning.

**Analysis on Size of Preference List**   To better understand the effect of *Preference List Ranking*, we conduct analysis on multiple choices of list sizes of TPO, and evaluate TPO on the ASDiv and GSM-Plus datasets using Qwen2-7B-Instruct. As illustrated in Fig. 3(b), as the size of the *Preference List Ranking* increases, the performance of TPO shows a steady growth, which is observed across both datasets. We argue that it is beneficial to model preferences using preference lists with varying reward values, and further increasing the size of *Preference List Ranking* can enhance the performance of preference learning.

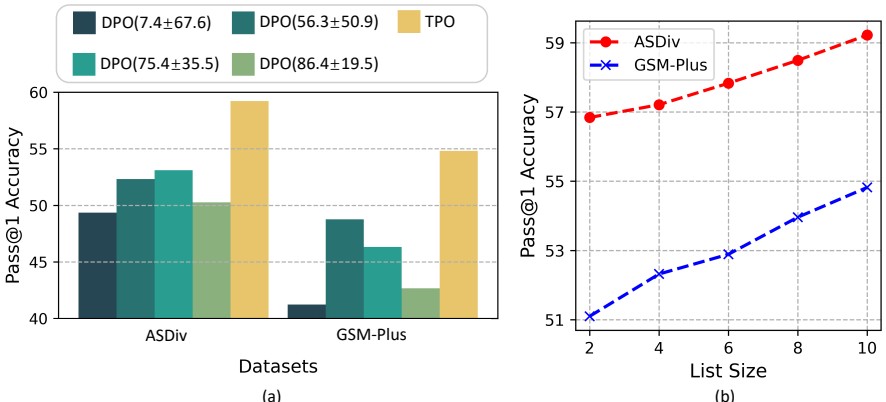

Figure 3: (a) illustrates a comparison between `TPO` and DPO using various reward value distributions for dispreferred responses on the ASDiv and GSM-Plus datasets. The numbers in the legend following each group of DPO algorithms represent the mean and standard deviation of the reward values for dispreferred responses. The results indicate that `TPO` consistently outperforms DPO. (b) shows performance of `TPO` with different list sizes on the ASDiv and GSM-Plus datasets. `TPO` benefits more and monotonically as the list size increases.

## 5 LIMITATIONS AND FUTURE WORKS

Despite the promising results obtained in our work, it is important to acknowledge the limitations. The first limitation is that `TPO` may introduce a stronger form of "catastrophic forgetting". The results in Sec. 4.2 indicate that while `TPO` exhibits excellent performance on in-distribution datasets, it may suffer from performance degradation on out-of-distribution datasets. We provide a more in-depth discussion in Sec. 4.2 and attribute this issue to "catastrophic forgetting" (Xuhong et al., 2018; Liao et al., 2022; 2024b). Existing strategies to mitigate "catastrophic forgetting" include memory replay (Miao et al., 2024; Babakniya et al., 2024), regularization constraints (Liao et al., 2022; 2024b), and meta-learning (Gupta et al., 2020; Son et al., 2024), among others. Incorporating these techniques into the `TPO` training procedure could potentially improve the generalization of `TPO` on out-of-distribution datasets.

The second limitation is due to the imbalanced distribution of the preference tree reward values, as shown in Fig. 2(c). We analyze the reasons for this as follows: (1) Autoregressive LLMs, including ChatGPT, tend to assign either high or low values (Xiong et al., 2023). Although we employ the Re-ACT strategy to prompt ChatGPT to provide a more reasonable evaluation, this issue remains to be addressed. (2) Our data generation pipeline, as depicted in Fig. 2(a), adopts a strategy of generating additional responses starting from correct trajectories. This strategy ensures that the preference tree contains at least one preferred response and that the reasoning generated from intermediate nodes includes some correct reasoning paths to diversify the reward values. However, once the preceding trajectory in the generated path already includes the key steps to solve the problem, the subsequent steps become easily inferable, leading to a higher distribution of reward values in the preference tree. In future work, we aim to introduce more effective ToT strategies, such as MCTS (Xie et al., 2024), to ensure the generation of higher-quality data. Additionally, we will employ techniques such as prompt optimization (Shin et al., 2020), multi-model collaborative scoring (Talebirad & Nadiri, 2023), and self-consistency (Wang et al., 2023) to enhance the reliability of the scoring procedure.

## 6 CONCLUSIONS

In this work, we propose `TPO`, a preference learning algorithm designed specifically for preference trees as input. `TPO` enhances DPO by addressing two critical issues: (1) DPO only supports binary preference input and cannot model preferences from preference lists with varying reward values. (2) DPO exhibits a lower reward margin when dealing with reasoning involving long chains of trajectories with shared sub-trajectories. We evaluate the effectiveness of `TPO` on extensive experiments, and the experimental results indicate that `TPO` consistently outperforms DPO on in-distribution data and shows promise for its generalization to out-of-distribution data.

## ACKNOWLEDGMENTS

This work was supported by the National Natural Science Foundation of China (Grant No. U23A20468). (The authors would like to express their sincere gratitude to all individuals and institutions who contributed to this research. Their valuable support and constructive suggestions greatly facilitated the completion of this study.)

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

APPENDIX

## A   ALGORITHM DESCRIPTION OF TPO

---

**Algorithm 1:** TPO Training Algorithm

---

**Input**   : Dataset $\mathcal{D} = (x^{(i)}, \mathbf{y}^{(i)}, \mathbf{v}^{(i)})_{i=1}^{M}$, Policy Model $\pi$
**Output:** Aligned Policy Model $\pi_\theta$

1  *Initialize $\pi_\theta$ and $\pi_{ref}$ with $\pi$ ;*
2  **for** $(x, \mathbf{y}, \mathbf{v}) \in \mathcal{D}$ **do**
     // Preference List Ranking
3     **for** $y_i, y_j \in \mathbf{y} \mid v_i > v_j$ **do**
          // Calculate the Lambda weight
4        $\lambda_{i,j} \leftarrow$ Eq. 8 ;
          // Adaptive Step Reward
5        **for** $s_k^{(i)} \in y_i, s_k^{(j)} \in y_j$ **do**
             // Adjust reward margin based on semantic similarity between
                paired steps
6           $\mathcal{RM}_{(i,j)} \leftarrow$ Eq. 10 ;

          // Calculate the preference list ranking loss
7        $\mathcal{L}_{PRL} \leftarrow$ Eq. 7 $\leftarrow \mathcal{RM}_{(i,j)} = r_i - r_j$ ;
8     Update Policy Model: $\pi_\theta \leftarrow \pi_\theta + \nabla\mathcal{L}_{PRL}$ ;
9  Return $\pi_\theta$.

---

## B   PROMPTS USED IN THIS WORK

**Prompt used for data generation.**   We employ the following prompts to synthesize the relevant data for preference trees. To ensure that the generated trajectories contain some correct reasoning steps, we provide the initial few reasoning steps in the prompts and allow LLMs to generate the subsequent reasoning steps.

---

Prompt used for data generation.

[System]
You are a helpful assistant.

[Instructions]
### Given the question, please provide the steps to solve it.
### Question: {question}

### Your answer should strictly follow the following format.
Step 1:
Step 2:
Step 3:
...

### Please reason step by step, and put your final answer within boxed{Your Answer}.
Step 1: {step_1}
Step 2: {step_2}
Step 3:

---

**Prompt used for generating reasoning trajectory scores using ChatGPT.**   We utilize the following prompts to instruct ChatGPT to score the reasoning trajectories within preference trees. To

ensure the reliability of the scores, we provide ChatGPT with genuine reasoning trajectories as a reference and employ the ReACT to facilitate ChatGPT in generating the scoring rationale.

---

Prompt used for generating reasoning trajectory scores using ChatGPT.

[System]
You are a helpful assistant.

[Instructions]
### Given the question, standard answer, and current answer, give a score for the current answer.
### Question: {question}
### Standard Answer: {standard_answer}
### Current Answer: {current_answer}

### You only need to give the score, and you also need to provide a detailed comparison with the standard answer to give the reason for your score.
### Provide a reward score between -100 and 100 for the answer quality, using very strict standards. Do not give a full score above 95. Make sure the reward score is an integer.
### If the final answer of the current answer is incorrect, please give a lower score.
### Your answer should strictly follow the following json format. Please note that only the following JSON is provided and no additional response content is required.
{
    "reasoning": "",
    "score": ""
}

### Your Answer:

---

**Prompt used for solving Math problems.**   In our assessment of `TPO` performance, we employ the following prompts to address relevant Math tasks, including MATH (Hendrycks et al., 2021), SVAMP (Patel et al., 2021), ASDiv (Miao et al., 2021) and GSM-Plus (Li et al., 2024a) datasets.

---

Prompt used for solving Math problems.

[Instructions]
Solve the following math problem step-by-step.
Simplify your answer as much as possible. Present your final answer as within boxed{Your Answer}
{question}

---

**Prompt used for solving Coding problems on HumanEval dataset.**   In our assessment of `TPO` performance, we employ the following prompts to address relevant Coding task on HumanEval (Chen et al., 2021) dataset.

---

Prompt used for solving Coding problems on HumanEval dataset.

[Instructions]
Write Python code to solve the task.
Write a Python function to solve the following problem: Present code in "'python"'
"'python
{question}
"'

---

**Prompt used for solving Coding problems on MBPP dataset.** In our assessment of `TPO` performance, we employ the following prompts to address relevant Coding task on MBPP (Austin et al., 2021) dataset.

> Prompt used for solving Coding problems on MBPP dataset.
>
> [Instructions]
> Write Python code to solve the task.
> Write a Python function to solve the following problem: Present code in "'python"'
> "'python
> {question}
> >>> {test_case}
> "'

**Prompt used for solving Reasoning problems on BBH dataset.** In our assessment of `TPO` performance, we employ the following prompts to address relevant Reasoning task on BBH (Suzgun et al., 2023) dataset.

> Prompt used for solving Reasoning problems on BBH dataset.
>
> [Instructions]
> Answer the following question.
> {question}
> Let's think step by step.

**Prompt used for solving Reasoning problems on MMLU dataset.** In our assessment of `TPO` performance, we employ the following prompts to address relevant Reasoning task on MMLU (Hendrycks et al.) dataset.

> Prompt used for solving Reasoning problems on MMLU dataset.
>
> [Instructions]
> Please answer the following multiple-choice questions.
> {question}
> Answer:

## C   ADDITIONAL EXPERIMENTAL RESULTS

### C.1   ADDITIONAL EXPERIMENTAL RESULTS ON MATH TASKS.

Table. 4 presents the performance of the `TPO` algorithm on mathematical tasks using Meta-Llama-3-8B-Instruct and Mistral-7B-Instruct-v0.3 as backbone LLMs. The experimental results demonstrate that `TPO` consistently outperforms the existing baseline algorithms across different backbone LLMs.

### C.2   COMPARISON WITH ADVANCED DPOS

Considering that `TPO` introduces potential data augmentation, we further introduce two new DPO baselines that extract complete pairwise data from the preference lists.

given data $[x, y_1, y_2, ..., y_n]$ with the preference ranking $y_1 \succ y_2 \succ, ..., \succ y_n$,

- **DPO**$^*$: DPO$^*$ extracts paired preferences $[x, (y_1, y_2), (y_1, y_3), ..., (y_1, y_n)]$.
- **DPO**$^+$: DPO$^+$ extracts paired preferences $[x, (y_1, y_2), (y_2, y_3), ..., (y_{n-1}, y_n)]$.

Table 4: Experimental results on **Math** task (In-Distribution) using Meta-Llama-3-8B-Instruct and Mistral-7B-Instruct-v0.3 as backbone LLMs. The best results for each large language model setting are indicated in **bold**. We report results using pass@1 accuracy.

| LLMs | size | open | general | MATH | SVAMP | ASDiv | GSM-Plus | Avg. |
|------|------|------|---------|------|-------|-------|----------|------|
| Meta-Llama-3-8B-Instruct | 8B | ✓ | ✓ | 28.50 | 74.30 | 75.57 | 48.77 | 56.79 |
| Meta-Llama-3-8B-Instruct+SFT | 8B | ✓ | ✓ | 28.08 | 73.30 | 74.92 | 47.50 | 55.95 |
| Meta-Llama-3-8B-Instruct+DPO | 8B | ✓ | ✓ | 29.12 | 73.10 | 75.73 | 47.66 | 56.40 |
| Meta-Llama-3-8B-Instruct+TPO | 8B | ✓ | ✓ | **29.58** | **77.70** | **80.91** | **53.25** | **60.36** |
| Mistral-7B-Instruct-v0.3 | 7B | ✓ | ✓ | 14.30 | 53.60 | 60.36 | 30.97 | 39.81 |
| Mistral-7B-Instruct-v0.3+SFT | 7B | ✓ | ✓ | 14.62 | 56.00 | 62.46 | 33.60 | 41.67 |
| Mistral-7B-Instruct-v0.3+DPO | 7B | ✓ | ✓ | 15.98 | 70.44 | 71.85 | 38.62 | 49.22 |
| Mistral-7B-Instruct-v0.3+TPO | 7B | ✓ | ✓ | **17.56** | **72.90** | **75.24** | **41.59** | **51.82** |

Table 5: Experimental results on **Math** task (In-Distribution) using the DPO, DPO*, DPO$^+$ and TPO algorithms. The best results for each large language model setting are indicated in **bold**. We report results using pass@1 accuracy.

| Methods | MATH | SVAMP | ASDiv | GSM-Plus | Avg. |
|---------|------|-------|-------|----------|------|
| Qwen2-7B-Instruct+DPO | 54.26 | 44.69 | 54.32 | 50.28 | 50.89 |
| Qwen2-7B-Instruct+DPO* | 55.02 | 46.85 | 57.96 | 53.26 | 53.27 |
| Qwen2-7B-Instruct+DPO$^+$ | 54.78 | 46.13 | 55.30 | 50.31 | 51.63 |
| Qwen2-7B-Instruct+TPO | **55.46** | **48.20** | **59.22** | **54.82** | **54.43** |
| DeepSeekMath-7B-Instruct+DPO | 48.66 | 85.98 | 90.16 | 62.86 | 71.92 |
| DeepSeekMath-7B-Instruct+DPO* | 50.33 | 86.20 | **90.61** | 63.57 | 72.68 |
| DeepSeekMath-7B-Instruct+DPO$^+$ | 49.75 | 85.78 | 89.78 | 63.11 | 72.11 |
| DeepSeekMath-7B-Instruct+TPO | **51.30** | **86.80** | **90.61** | **64.73** | **73.36** |

We conducted experiments on Qwen2-7B-Instruct and DeepSeekMath-7B-Instruct, and the experimental results are shown in the Table. 5. The experimental results indicate that TPO outperforms all the baseline models. We argue that all variants of DPO focus solely on the relative likelihood between chosen and rejected preferences. However, this approach causes the likelihood of the chosen preferences to decrease during the optimization process (Chen et al., 2024a; Pal et al., 2024). In contrast, TPO introduces lambda weights into the ranking algorithm to provide absolute positional information for preferences within the list, mitigating data likelihood decline issues.

## C.3 Analysis of Various Ranking Losses

We further introduced various ranking losses, including Pointwise SoftMax (Cao et al., 2007), Pairwise Logistic (Burges et al., 2005), and Listwise MLE (Xia et al., 2008), to further confirm the validity of choosing LambdaLoss for TPO. The experimental results are shown in Table. 6:

We have the following observations: Pointwise SoftMax demonstrates the poorest performance, indicating that learning only the reward values of preferences is insufficient. The relative relationships between preferences are particularly important. Pairwise Logistic and Listwise MLE perform worse than LambdaLoss. This is because both approaches only consider the relative relationships between preferences while neglecting their absolute positions in the ranked list. This limitation has been shown in existing Learn-to-Rank literature (Wang et al., 2018; Jagerman et al., 2022) to be detrimental to ranking optimization. It also underscores the motivation for introducing lambda weights in this work.

Table 6: Performance of `TPO` using various ranking losses on **Math** tasks. The best results for each large language model setting are indicated in **bold**. We report results using pass@1 accuracy.

| Ranking Loss | MATH | SVAMP | ASDiv | GSM-Plus | Avg. |
|---|---|---|---|---|---|
| Qwen2-7B-Instruct | | | | | |
| Pointwise SoftMax | 52.13 | 43.22 | 52.16 | 48.77 | 49.07 |
| Pairwise Logistic | 55.02 | 46.85 | 57.96 | 53.26 | 53.27 |
| Listwise MLE | 54.10 | 45.89 | 54.16 | 52.33 | 51.62 |
| LambdaLoss | **55.46** | **48.20** | **59.22** | **54.82** | **54.43** |
| DeepSeekMath-7B-Instruct | | | | | |
| Pointwise SoftMax | 48.59 | 84.26 | 87.89 | 60.96 | 70.43 |
| Pairwise Logistic | 50.33 | 86.20 | 90.51 | 63.57 | 72.65 |
| Listwise MLE | 50.10 | 85.74 | 89.22 | 62.41 | 71.87 |
| LambdaLoss | **51.30** | **86.80** | **90.61** | **64.73** | **73.36** |

Table 7: The t-test experimental results for analyzing the effectiveness of *Adaptive Step Reward* are reported, including the mean value and standard error of pass@1 accuracy. ‡ denotes $p < 0.01$, and † denotes $p < 0.05$.

| Methods | MATH | SVAMP | ASDiv | GSM-Plus |
|---|---|---|---|---|
| Qwen2-7B-Instruct | | | | |
| TPO | $55.54 \pm 0.06^{\ddagger}$ | $48.00 \pm 0.06^{\dagger}$ | $59.25 \pm 0.11^{\ddagger}$ | $54.86 \pm 0.04^{\ddagger}$ |
| w/o Adaptive Step Reward | $55.18 \pm 0.07$ | $47.80 \pm 0.07$ | $58.60 \pm 0.15$ | $54.33 \pm 0.05$ |
| p-value | 0.0004 | 0.0361 | 0.0013 | 1.0970e-9 |
| DeepSeekMath-7B-Instruct | | | | |
| TPO | $51.31 \pm 0.09^{\ddagger}$ | $86.85 \pm 0.07^{\dagger}$ | $90.83 \pm 0.11^{\dagger}$ | $64.67 \pm 0.04^{\ddagger}$ |
| w/o Adaptive Step Reward | $50.78 \pm 0.08$ | $86.55 \pm 0.10$ | $90.46 \pm 0.08$ | $64.34 \pm 0.04$ |
| p-value | 9.9973e-5 | 0.0167 | 0.0129 | 2.8655e-6 |

## C.4 VALIDATION OF ADAPTIVE STEP REWARD EFFECTIVENESS

To assess the statistical significance of our proposed *Adaptive Step Reward*, we performed 20 independent experiments for both `TPO` and `TPO` w/o *Adaptive Step Reward* under identical experimental conditions. The performance metrics from these experiments were then analyzed using an independent t-test to compare the means and the standard error of the two groups. The experimental results are shown in the Table. 7.

The null hypothesis assumed no significant difference in performance between `TPO` and `TPO` w/o *Adaptive Step Reward*. A p-value less than 0.01/0.05 was considered statistically significant, indicating that the observed differences were unlikely to occur by chance. The results showed that `TPO` consistently outperformed the `TPO` w/o *Adaptive Step Reward*, with statistically significant improvements in pass@1 accuracy, validating the effectiveness of *Adaptive Step Reward*.

## C.5 CASE STUDIES

To better understand the motivation behind the *Adaptive Step Reward*, we provide two cases for further analysis below.

---

$Case_1$. The critical errors in the steps are highlighted in red font.

**Prompt:**
Let $A = 2 + i$, $B = 3 - 2i$, $O = -4$, $P = -i$, $Q = 5 + 3i$, $S = 5 - 3i$, and $T = -3 - 6i$.
Calculate $(A + B - O + P + Q * S) + T$.
Let's think step by step.
Step 1:
First, we need to find the sum of all the complex numbers in the problem.
We are given $A = 2 + i$, $B = 3 - 2i$, $O = -4$, $P = -i$, $Q = 5 + 3i$, $S = 5 - 3i$, and $T = -3 - 6i$.
We need to find $(A + B - O + P + Q * S) + T$.
Step 2:

**$Response_1$**
Let's add all the complex numbers together.
First, add $A$ and $B$:
$A + B = (2 + i) + (3 - 2i) = 2 + 3 + i - 2i = 5 - i$
Then, subtract $O$:
$5 - i - (-4) = 5 - i + 4 = 9 - i$
Next, add $P$:
$9 - i + (-i) = 9 - 2i$
Then, calculate $Q * S$:
$(5 + 3i) * (5 - 3i) = 25 - 15i + 15i - 9 = 16$
Next, add $Q * S$:
$9 - 2i + 16 = 25 - 2i$
Finally, add $T$:
$25 - 2i + (-3 - 6i) = 25 - 2i - 3 - 6i = 22 - 8i$
So, $(A + B - O + P + Q + S) + T = 22 - 8i$.

**$Response_2$**
First, let's add all the given complex numbers:
$A + B = (2 + i) + (3 - 2i) = 5 - i$,
$-O = -(-4) = 4$,
$P = -i$,
$Q * S = (5 + 3i) * (5 - 3i) = 25 + 15i - 15i - 9 = 16$,
$T = -3 - 6i$.
Now, let's add these results together:
$5 - i + 4 - i + 16 - 3 - 6i = 22 - 8i$.

**$Response_3$**
Let's add all the complex numbers together.
First, add $A$ and $B$:
$A + B = (2 + i) + (3 - 2i) = 2 + 3 + i - 2i = 5 - i$
Then, subtract $O$:
$5 - i - (-4) = 5 - i + 4 = 9 - i$
Next, add $P$:
$9 - i + (-i) = 9 - 2i$
Then, add $Q$:
$9 - 2i + (5 + 3i) = 9 - 2i + 5 + 3i = 14 + i$
Next, multiply $S$:
$(14 + i) * (5 - 3i) = 70 + 5i - 42i - 3 = 67 - 37i$
 Finally, add $T$:
$67 - 37i + (-3 - 6i) = 67 - 37i - 3 - 6i = 64 - 43i$
So, $(A + B - O + P + Q * S) + T = 64 - 43i$.

$Case_2$. The critical errors in the steps are highlighted in red font.

**Prompt:**
If Jeremy played a three-letter word on a triple word score to earn thirty points, and before the word score was tripled, the first and third letters were each worth one point, what was the value of the middle letter before the word score was tripled?
Let's think step by step.
Step 1: The word is a three-letter word, and the first and third letters were each worth one point, so the total value of the first and third letters before the word score was tripled is 1+1 = 2 points.
Step 2:

**$Response_1$**
The word score was tripled to earn 30 points. So, the original word score before tripling was 30/3 = 10 points.

**$Response_2$**
The total score of the word before applying the triple word multiplier was calculated as 30/3 = 10, representing the base score.

**$Response_3$**
The word score was tripled to earn 30 points. So, the value of the middle letter before the word score was tripled is 30 - 2 = 28 points.

It is noteworthy that, in $Case_1$, $Responses_1$ and $Responses_2$, despite having significantly different expressions, both follow the correct steps, whereas $Response_3$, though similar to $Response_1$ in expression, fails to prioritize multiplication and thus yields an incorrect result. In the ground truth ranking, the preference ranking is $Response_1 \succ Response_2 \succ Response_3$. However, during LLMs' preference learning, since implicit reward are computed token by token, the significant overlap of tokens between $Response_3$ and $Response_1$ leads to an incorrect ranking of $Response_1 \succ Response_3 \succ Response_2$. A similar phenomenon can also be observed in $Case_2$.

Based on this observation, we propose the *Adaptive Step Reward* mechanism leveraging step-level semantic similarity to adjust the reward margin between preferences. We emphasize the importance of semantics in preference ranking and believe this is particularly critical in tasks like mathematical reasoning, which emphasize semantic similarity rather than token overlap.

We visualized the reward margins during the training process for the two cases in Fig. 4. The experimental results indicate that when the *Adaptive Step Reward* in TPO is disabled, LLMs exhibit a smaller reward margin between $Responses_1$ and $Responses_3$, which share a high degree of token overlap. Conversely, when *Adaptive Step Reward* is enabled, LLMs demonstrate a smaller reward margin between $Responses_1$ and $Responses_2$, which share similar semantics. This confirms the effectiveness of the *Adaptive Step Reward*, as it encourages preference optimization to focus more on semantic information rather than token overlap or sequential order.

## C.6  ANALYSIS OF $\beta$ IN TPO

We further conducted additional experiments to investigate the impact of $\beta$ on the performance of DPO and TPO. We used Qwen2-7B-Instruct as backbone LLM. The experimental results are shown in Table. 8. The results indicate that reducing $\beta$ in DPO from 0.5 to 0.1 or even 0.01 improves the performance of LLMs. However, it is noteworthy that TPO's performance also improves and consistently outperforms DPO.

In DPO, $\beta$ is a parameter that controls the deviation from the base reference policy. Therefore, when a smaller $\beta$ is used, it means that the constraints between the policy and the reference policy are relaxed, making it easier for the policy to adapt to the training task, leading to improved performance. This holds true for TPO as well. However, exist work (Wu et al., 2024) show that smaller values of $\beta$ are always better, as excessive deviation from the reference policy can lead to more forgetting,

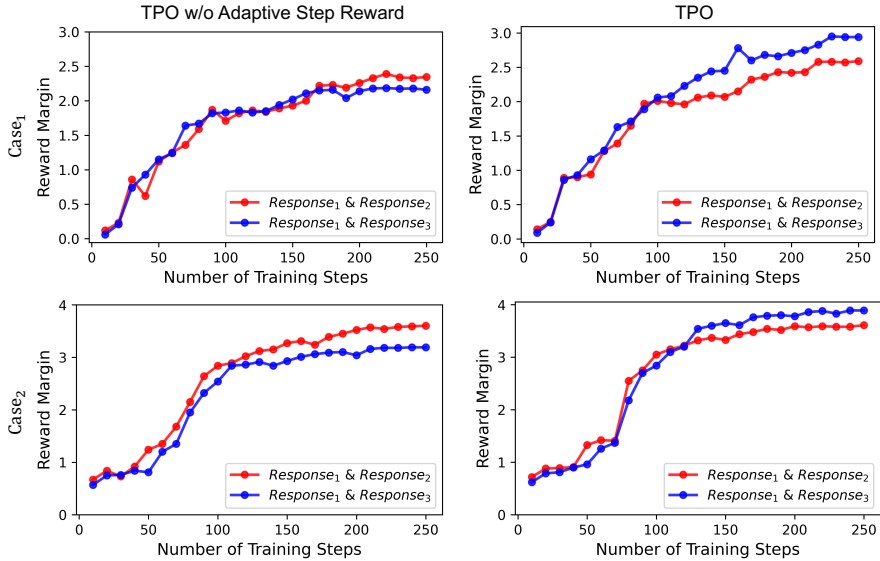

Figure 4: Study of Reward Margins.

Table 8: Experimental results of DPO and `TPO` across different $\beta$ settings on **Math** task. The better results for each large language model setting are indicated in **bold**. We report results using pass@1 accuracy.

| Methods | MATH | SVAMP | ASDiv | GSM-Plus | Avg. |
|---|---|---|---|---|---|
| | | $beta$=0.5 | | | |
| DPO | 54.26 | 44.69 | 54.32 | 50.28 | 50.89 |
| TPO | **55.46** | **48.20** | **59.22** | **54.82** | **54.43** |
| | | $beta$=0.1 | | | |
| DPO | 54.94 | 45.10 | 55.02 | 50.89 | 51.49 |
| TPO | **56.16** | **48.50** | **59.54** | **55.36** | **54.89** |
| | | $beta$=0.01 | | | |
| DPO | 55.30 | 45.30 | 55.34 | 51.19 | 51.78 |
| TPO | **56.48** | **48.50** | **60.03** | **55.54** | **55.14** |

which may cause a loss in performance on other tasks. Thus, we view the choice of $\beta$ as a trade-off between optimizing LLM performance on specific tasks and ensuring generalization across tasks.

## D   DISCUSSION ON RELATED LITERATURE

There are several contemporaneous literature (Chen et al., 2024b; Liao et al., 2024a; Scheid et al., 2024) with `TPO` that also analyze issues related to preference ranking in lists. We present the optimization objectives of `TPO` and related literature (Chen et al., 2024b; Liao et al., 2024a; Scheid et al., 2024) in Table. 9.

S-DPO (Chen et al., 2024b) employs SoftMax to maximize the reward margin between positive preference and all negative preferences. However, S-DPO does not account for the contrasts between negative preferences, nor does it consider the absolute positional information of preferences within the list. Although Rose-DPO (Liao et al., 2024a) does not consider the ranking of the preference list, similar to `TPO`, both Rose-DPO and `TPO` use additional weights to adjust the reward margin. The difference lies in the adjustment mechanism: while Rose-DPO is based on uncertainty, `TPO` adjusts

Table 9: Optimization Objectives for Preference Alignment. Where $\epsilon_\phi$ used in Rose-DPO represents the uncertainty assessment of preferences.

| Methods | Objective |
|---|---|
| S-DPO (Chen et al., 2024b) | $-\log \sigma \left( -\log \sum_{y_l \in \mathcal{Y}_l} \exp \left( \beta \log \frac{\pi_\theta(y_w|x)}{\pi_{\text{ref}}(y_w|x)} - \beta \log \frac{\pi_\theta(y_l|x)}{\pi_{\text{ref}}(y_l|x)} \right) \right)$ |
| Rose DPO (Liao et al., 2024a) | $-(1 - \epsilon_\phi) \left( \beta \log \frac{\pi_\theta(y_w|x)}{\pi_{\text{ref}}(y_w|x)} - \beta \log \frac{\pi_\theta(y_l|x)}{\pi_{\text{ref}}(y_l|x)} \right)$ $-\epsilon_\phi \left( \beta \log \frac{\pi_\theta(y_l|x)}{\pi_{\text{ref}}(y_l|x)} - \beta \log \frac{\pi_\theta(y_w|x)}{\pi_{\text{ref}}(y_w|x)} \right)$ |
| ODPO (Scheid et al., 2024) | $-\log \prod_{i=1}^{N} \frac{\exp(\beta \log \frac{\pi_\theta(y_i|x)}{\pi_{\text{ref}}(y_i|x)})}{\sum_{j=i}^{N} \exp(\beta \log \frac{\pi_\theta(y_j|x)}{\pi_{\text{ref}}(y_j|x)})}$ |
| TPO | $-\lambda_{i,j} \sum_{v_i > v_j} \log \sigma(\beta \log \frac{\pi_\theta(y_i|x)}{\pi_{\text{ref}}(y_i|x)} - \beta \log \frac{\pi_\theta(y_j|x)}{\pi_{\text{ref}}(y_j|x)})$ |

the margin according to the semantic similarity between preference pairs. ODPO (Scheid et al., 2024) utilizes Maximum Likelihood Estimation (MLE) to optimize list-wise preferences. However, existing Learn-to-Rank literature (Wang et al., 2018; Jagerman et al., 2022) suggests that list-MLE is not an ideal ranking optimization objective, as it enforces strict list ordering without considering the actual label values.

