# OpenReview forum: "TPO: Aligning Large Language Models with Multi-branch & Multi-step Preference Trees"
_ICLR.cc/2025/Conference — ICLR 2025 Poster_

### Official Review · Reviewer_Svzf · 2024-10-28

**Soundness:** 3
**Presentation:** 3
**Contribution:** 2
**Rating:** 6
**Confidence:** 4

**Summary:**

This paper presents Tree Preference Optimization (TPO), a novel framework that decomposes Tree of Thoughts into multi-branch and multi-step response patterns for fine-grained preference optimization. The framework reconceptualizes language model alignment as a generalized Preference List Ranking problem, which facilitates more effective alignment learning from preference lists. TPO introduces the Adaptive Step Reward mechanism, which adjusts step-wise rewards based on inter-step correlations to focus on discriminative steps, thereby enhancing the optimization efficiency.

**Strengths:**

- The paper is well-written and presents comprehensive empirical results with publicly available implementation code, demonstrating the effectiveness of the proposed approach.

- This paper presents a novel approach by decomposing Tree of Thoughts into a hierarchical structure of multi-branch, multi-step responses and leveraging Learning-to-Rank algorithms for preference optimization.

**Weaknesses:**

- The rationale for selecting LambdaLoss over other LTR loss functions in Section 3.1 requires further clarification. Maybe you can justify this choice by highlighting the specific advantages of LambdaLoss in your application.


- In Equation 10, I notice that RM=0 in Line 298, while my understanding suggests that cosine similarity should be 1 when the content is shared. Could you please clarify this point?  Maybe you can provide more explanation of your *Adaptive Step Reward*.

- The ablation studies for Adaptive Step Reward appear limited. Based on Table 3, its impact seems modest. Further investigation into the effectiveness and mechanism of Adaptive Step Reward would strengthen the analysis.

**Questions:**

See weaknesses

---

> ### Author Response · Authors · 2024-11-22
> **Response to Reviewer Svzf (1/3)**
>
> Thanks a lot for your acknowledgment of the writing, presentation, effectiveness and novelty of our submission!
>
> **Response to W1** about the reasons for choosing LambdaLoss.
>
> Thank you for your valuable suggestion in emphasizing the advantages of LambdaLoss. As stated in Line 230 of the original manuscript, we chose LambdaLoss because it emphasizes the absolute position of preferences in the list, which helps further guide LLMs in aligning preferences.
> - We have also introduced additional ranking losses, including Pointwise SoftMax [1], Pairwise Logistic [2], and Listwise MLE [3], to further confirm the validity of choosing LambdaLoss for TPO. The experimental results are shown in the table below:
>
> | Ranking Loss             | MATH  | SVAMP | ASDiv | GSM-Plus | Avg.  |
> |--------------------------|-------|-------|-------|----------|-------|
> | Qwen2-7B-Instruct        |       |       |       |          |       |
> | Pointwise SoftMax        | 52.13 | 43.22 | 52.16 | 48.77    | 49.07 |
> | Pairwise Logistic        | 55.02 | 46.85 | 57.96 | 53.26    | 53.27 |
> | Listwise MLE             | 54.10 | 45.89 | 54.16 | 52.33    | 51.62 |
> | LambdaLoss (Ours)              | **55.46** | **48.20** | **59.22** | **54.82**    | **54.43** |
> | DeepSeekMath-7B-Instruct |       |       |       |          |       |
> | Pointwise SoftMax        | 48.59 | 84.26 | 87.89 | 60.96    | 70.43 |
> | Pairwise Logistic        | 50.33 | 86.20 | 90.51 | 63.57    | 72.65 |
> | Listwise MLE             | 50.10 | 85.74 | 89.22 | 62.41    | 71.87 |
> | LambdaLoss (Ours)              | **51.30** | **86.80** | **90.61** | **64.73**    | **73.36** |
>
> We observe the following:
>
> - Pointwise SoftMax demonstrates the poorest performance, indicating that learning only the reward values of preferences is insufficient. The relative relationships between preferences are particularly important.
>
> - Pairwise Logistic and Listwise MLE perform worse than LambdaLoss. This is because both approaches only consider the relative relationships between preferences while neglecting their absolute positions in the ranked list. This limitation has been shown in existing Learn-to-Rank literature [4,5] to be detrimental to ranking optimization. It also underscores the motivation for introducing lambda weights in this work (see Line 230 of the original manuscript).
>
>
> **Response to W2** about the clarification on Equation 10.
> - In Line 298 of the manuscript, the reason for $\mathcal{RM}=0$ is that $\beta\log\frac{\pi _ \theta(s^{i} _ k\mid x)}{\pi _ {\mathrm{ref}}(s^{i} _ k\mid x)}-\beta\log\frac{\pi _ \theta(s^{j} _ k\mid x)}{\pi _ {\mathrm{ref}}(s^{j} _ k\mid x)}=0$, not $1+\frac{emb(s^i _ k)\cdot emb(s^j _ k)}{\|emb(s^i _ k)\|\|emb(s^j _ k)\|}=0$. We have revised the manuscript to clarify this description.

---

> ### Author Response · Authors · 2024-11-22
> **Response to Reviewer Svzf (2/3)**
>
> **Response to W2 and W3** about the analysis of *Adaptive Step Reward*. Thank you for your suggestions. We are pleased to discuss our further experiments and insights on the *Adaptive Step Reward* with you.
>
> - *T-test on Adaptive Step Reward*. To assess the statistical significance of our proposed *Adaptive Step Reward*, we performed 20 independent experiments for both TPO and TPO w/o *Adaptive Step Reward* under identical experimental conditions. The performance metrics from these experiments were then analyzed using an independent t-test to compare the means and the standard error of the two groups. The experimental results are shown in the table below, where $\^{*}$ denotes $p<0.01$, and $\^{+}$ denotes $p<0.05$.
>
>
> | Methods                  | MATH       | SVAMP      | ASDiv      | GSM-Plus   |
> |--------------------------|------------|------------|------------|------------|
> | Qwen2-7B-Instruct        |            |            |            |            |
> | TPO                      | 55.54±0.06$\^{*}$ | 48.00±0.06$\^{+}$ | 59.25±0.11$\^{*}$ | 54.86±0.04$\^{*}$ |
> | w/o Adaptive Step Reward | 55.18±0.07 | 47.80±0.07 | 58.60±0.15 | 54.33±0.05 |
> | p-value                  | 0.0004     | 0.0361     | 0.0013     | 1.0970e-9  |
> | DeepSeekMath-7B-Instruct |            |            |            |            |
> | TPO                      | 51.31±0.09$\^{*}$ | 86.85±0.07$\^{+}$ | 90.83±0.11$\^{+}$ | 64.67±0.04$\^{*}$ |
> | w/o Adaptive Step Reward | 50.78±0.08 | 86.55±0.10 | 90.46±0.08 | 64.34±0.04 |
> | p-value                  | 9.9973e-5  | 0.0167     | 0.0129     | 2.8655e-6  |
>
> The null hypothesis assumed no significant difference in performance between TPO and TPO w/o *Adaptive Step Reward*. A p-value less than 0.01/0.05 was considered statistically significant, indicating that the observed differences were unlikely to occur by chance. The results showed that **TPO consistently outperformed the TPO w/o *Adaptive Step Reward***, with statistically significant improvements in pass@1 accuracy, validating the effectiveness of *Adaptive Step Reward*.
>
> - *Case study on Adaptive Step Reward*. We further clarify our inspiration for *Adaptive Step Reward* through a case study. The following case is provided:
>
> ```
> **Prompt:**
> Let $A=2+i$, $B=3-2i$, $O=-4$, $P=-i$, $Q=5+3i$, $S=5-3i$, and $T=-3-6i$. Calculate $(A+B-O+P+Q*S)+T$.
> Let's think step by step.
> Step 1:
> First, we need to find the sum of all the complex numbers in the problem.
> We are given $A=2+i$, $B=3-2i$, $O=-4$, $P=-i$, $Q=5+3i$, $S=5-3i$, and $T=-3-6i$.
> We need to find $(A+B-O+P+Q*S)+T$.
> Step 2:
>
> **Response_1**
> Let's add all the complex numbers together.
> First, add $A$ and $B$:
> $A+B = (2+i) + (3-2i) = 2+3+i-2i = 5-i$
> Then, subtract $O$:
> $5-i - (-4) = 5-i+4 = 9-i$
> Next, add $P$:
> $9-i + (-i) = 9-2i$
> Then, calculate $Q*S$:
> $(5+3i) * (5-3i) = 25-15i+15i-9 = 16$
> Next, add $Q*S$:
> $9-2i + 16 = 25-2i$
> Finally, add $T$:
> $25-2i + (-3-6i) = 25-2i-3-6i = 22-8i$
> So, $(A+B-O+P+Q+S)+T = 22-8i$.
>
> **Response_2**
> First, let's add all the given complex numbers:
> $A+B = (2+i) + (3-2i) = 5-i$,
> $-O = -(-4) = 4$,
> $P = -i$,
> $Q*S = (5+3i) * (5-3i) = 25+15i-15i-9 = 16$,
> $T = -3-6i$.
> Now, let's add these results together:
> $5 - i + 4 - i + 16 -3 -6i = 22 - 8i$.
>
> **Response_3**
> Let's add all the complex numbers together.
> First, add $A$ and $B$:
> $A+B = (2+i) + (3-2i) = 2+3+i-2i = 5-i$
> Then, subtract $O$:
> $5-i - (-4) = 5-i+4 = 9-i$
> Next, add $P$:
> $9-i + (-i) = 9-2i$
> Then, add $Q$:
> $9-2i + (5+3i) = 9-2i+5+3i = 14+i$
> Next, multiply $S$:
> $(14+i) * (5-3i) = 70+5i-42i-3 = 67-37i$
> Finally, add $T$:
> $67-37i + (-3-6i) = 67-37i-3-6i = 64-43i$
> So, $(A+B-O+P+Q*S)+T = 64-43i$.
> ```
>
> It is noteworthy that $Responses_1$ and $Responses_2$, despite having significantly different expressions, both follow the correct steps, whereas $Response_3$, though similar to $Response_1$ in expression, fails to prioritize multiplication and thus yields an incorrect result. In the ground truth ranking, the preference ranking is $Response_1 \succ Response_2 \succ Response_3$. However, during LLMs’ preference learning, since implicit reward are computed token by token, the significant overlap of tokens between $Response_3$ and $Response_1$ leads to an incorrect ranking of $Response_1 \succ Response_3 \succ Response_2$.
>
> Based on this observation, we propose the *Adaptive Step Reward* mechanism leveraging step-level semantic similarity to adjust the reward margin between preferences. **We emphasize the importance of semantics in preference ranking and believe this is particularly critical in tasks like mathematical reasoning, which emphasize semantic similarity rather than token overlap.**
>
> We sincerely invite you to review the analysis provided in **Line 1026 of the revised manuscript**, where we present additional case studies and demonstrate the success of the *Adaptive Step Reward* in these cases. These examples help to further elucidate the insights and effectiveness of our *Adaptive Step Reward* approach.

---

> ### Author Response · Authors · 2024-11-22
> **Response to Reviewer Svzf (3/3)**
>
> **References**
>
> [1] "Learning to rank: from pairwise approach to listwise approach." ICML (2007).
>
> [2] "Learning to rank using gradient descent." ICML (2005).
>
> [3] "Listwise approach to learning to rank: theory and algorithm." ICML (2008).
>
> [4] "The lambdaloss framework for ranking metric optimization." CIKM (2018).
>
> [5] "On optimizing top-k metrics for neural ranking models." SIGIR (2022).
>
> We have added the additional case study analysis and the comparison of different ranking losses to the revised manuscript. To help you quickly locate our modifications, we have highlighted the changes using stone blue and bright red fonts. (Please refer to the *Guideline for Reviewers* on the last page of the revised manuscript.)
>
>
> We want to express our sincere gratitude for your review. If you have any further questions or concerns, please feel free to contact us at any time. We are always available and look forward to further discussions with you.
>
> Best regards,
>
> All Authors

---

> ### Author Response · Authors · 2024-11-25
> **Kindly Request for Reviewer's Feedback**
>
> Dear Reviewer Svzf,
>
> We understand that you may be busy, and we sincerely appreciate your efforts in reviewing our paper. We have made significant efforts to address the concerns you raised during the initial review, but we have not yet received any feedback.
>
> As the rebuttal phase is nearing its end, we would like to kindly follow up. We would greatly appreciate it if you could share your thoughts on our responses. If there are still any issues or areas that require further improvement, we are more than willing to continue refining our work.
>
> Thank you again for your time and valuable feedback!

---

> > ### Comment · Reviewer_Svzf · 2024-11-26
> >
> > Thank you for your response and the additional experiments, which have addressed some of our concerns.
> >
> > Additionally, we have several questions. First, why is it necessary to use a list-based learning-to-rank approach? What would be the outcome if we simply divided the data into pairs and applied existing preference alignment methods such as SimPO or other DPO variants? Second, regarding the DPO comparison where beta was set to 0.5 - what was the rationale behind this choice? The original DPO paper used beta=0.1, and current practice suggests that beta=0.01 tends to yield better results.

---

> > > ### Author Response · Authors · 2024-11-28
> > > **Response to Reviewer Svzf (1/2)**
> > >
> > > Thank you for acknowledging our previous round of responses. We are pleased to receive further suggestions to improve the shortcomings in our original manuscript.
> > >
> > >
> > > **Response to additional Q1** about the comparison between TPO and DPO using paired data extracted from the list.
> > >
> > > We introduce two new baselines, which extract paired data from the preference list and applies the DPO algorithm. Specifically, given data $[x, y_1, y_2, ..., y_n]$ with the preference ranking $y_1 \succ y_2 \succ, ... , \succ y_n$, we extract two types of data:
> > > - $[x, (y_1, y_2), (y_1, y_3), ..., (y_1, y_n)]$ (denoted as $DPO^{*}$)
> > > - $[x, (y_1, y_2), (y_2, y_3), ..., (y_{n-1}, y_n)]$ (denoted as $DPO^{+}$).
> > >
> > > We conducted experiments on `Qwen2-7B-Instruct` and `DeepSeekMath-7B-Instruct`, and the experimental results are shown in the table below. The experimental results indicate that **TPO outperforms all the baseline algorithm**.
> > >
> > >
> > > | Methods                        | MATH  | SVAMP | ASDiv | GSM-Plus | Avg.  |
> > > |--------------------------------|-------|-------|-------|----------|-------|
> > > | Qwen2-7B-Instruct+DPO          | 54.26 | 44.69 | 54.32 | 50.28    | 50.89 |
> > > | Qwen2-7B-Instruct+$DPO^{*}$        | 55.02 | 46.85 | 57.96 | 53.26    | 53.27 |
> > > | Qwen2-7B-Instruct+$DPO^{+}$        | 54.78 | 46.13 | 55.30 | 50.31    | 51.63 |
> > > | Qwen2-7B-Instruct+TPO          | **55.46** | **48.20** | **59.22** | **54.82**    | **54.43** |
> > > | DeepSeekMath-7B-Instruct+DPO   | 48.66 | 85.98 | 90.16 | 62.86    | 71.92 |
> > > | DeepSeekMath-7B-Instruct+$DPO^{*}$ | 50.33 | 86.20 | **90.61** | 63.57    | 72.68 |
> > > | DeepSeekMath-7B-Instruct+$DPO^{+}$ | 49.75 | 85.78 | 89.78 | 63.11    | 72.11 |
> > > | DeepSeekMath-7B-Instruct+TPO   | **51.30** | **86.80** | **90.61** | **64.73**    | **73.36** |
> > >
> > > Our **perspective** is as follows:
> > > - We argue that all variants of DPO focus solely on the relative likelihood between chosen and rejected preferences. However, this approach causes the likelihood of the chosen preferences to decrease during the optimization process [1,2]. In contrast, TPO introduces lambda weights into the ranking algorithm to provide absolute positional information for preferences within the list, mitigating data likelihood decline issues.

---

> > > > ### Comment · Reviewer_Svzf · 2024-11-30
> > > >
> > > > The pair construction approach in this work differs from that of lambda loss, resulting in fewer training pairs for $DPO^{*}$ and $DPO^{+}$ compared to lambda loss.

---

> > > ### Author Response · Authors · 2024-11-28
> > > **Response to Reviewer Svzf (2/2)**
> > >
> > > **Response to additional Q2** about the $\beta$ in DPO.
> > >
> > > - *Reason Statement*: The reason we set the parameter $\beta=0.5$ in DPO is solely because we followed the experimental setup of [3] (which is cited in line 49 of our original manuscript). This work also uses DPO to optimize the performance of LLMs on long-chain reasoning tasks.
> > >
> > > - *Fairness Statement*: In our original manuscript, we set $\beta=0.5$ for **both DPO and TPO**, ensuring fairness in the comparison.
> > >
> > > - *Analysis of $\beta$*: We further conducted additional experiments to investigate the impact of $\beta$ on the performance of DPO and TPO. We used `Qwen2-7B-Instruct` as backbone LLM. The experimental results are shown in the table below. The results indicate that reducing $\beta$ in DPO from 0.5 to 0.1 or even 0.01 improves the performance of LLMs. However, it is noteworthy that TPO's performance also improves and consistently outperforms DPO.
> > >
> > >
> > > | Methods   | MATH  | SVAMP | ASDiv | GSM-Plus | Avg.  |
> > > |-----------|-------|-------|-------|----------|-------|
> > > | $\beta$=0.5  |       |       |       |          |       |
> > > | DPO       | 54.26 | 44.69 | 54.32 | 50.28    | 50.89 |
> > > | TPO       | **55.46** | **48.20**  | **59.22** | **54.82**    | **54.43** |
> > > | $\beta$=0.1  |       |       |       |          |       |
> > > | DPO       | 54.94 | 45.10  | 55.02 | 50.89    | 51.49 |
> > > | TPO       | **56.16** | **48.50**  | **59.54** | **55.36**    | **54.89** |
> > > | $\beta$=0.01 |       |       |       |          |       |
> > > | DPO       | 55.30  | 45.30  | 55.34 | 51.19    | 51.78 |
> > > | TPO       | **56.48** | **48.50**  | **60.03** | **55.54**    | **55.14** |
> > >
> > > Our **perspective** is as follows:
> > > - In DPO, $\beta$ is a parameter that controls the deviation from the base reference policy. Therefore, when a smaller $\beta$ is used, it means that the constraints between the policy and the reference policy are relaxed, making it easier for the policy to adapt to the training task, leading to improved performance. **This holds true for TPO as well**.
> > >
> > > - However, we do not believe that smaller values of $\beta$ are always better, as excessive deviation from the reference policy can lead to more forgetting, which may cause a loss in performance on other tasks. Thus, we view the choice of $\beta$ as a trade-off between optimizing LLM performance on specific tasks and ensuring generalization across tasks. [4] conducted a more detailed analysis of $\beta$ in DPO and reached conclusions similar to ours.
> > >
> > >
> > > **References**:
> > >
> > > [1] "Noise contrastive alignment of language models with explicit rewards." NeurIPS (2024).
> > >
> > > [2] "Smaug: Fixing failure modes of preference optimisation with dpo-positive." arXiv preprint arXiv:2402.13228 (2024).
> > >
> > > [3] "Step-dpo: Step-wise preference optimization for long-chain reasoning of llms." arXiv preprint arXiv:2406.18629 (2024).
> > >
> > > [4] "$\beta$-DPO: Direct Preference Optimization with Dynamic $\beta$." NeurIPS (2024).
> > >
> > >
> > > We have revised the manuscript to include the new experiments and discussions. Please refer to the last page of the revised manuscript for easy access to the relevant changes.
> > >
> > > We sincerely appreciate your suggestions, which are crucial to improving the quality of our manuscript. If you have any further questions, please feel free to reach out to us. We would be happy to discuss any issues related to TPO with you.

---

> > > > ### Comment · Reviewer_Svzf · 2024-11-30
> > > >
> > > > The authors have adequately addressed most of our concerns. Based on their responses, we will increase our rating accordingly. We appreciate the authors' thorough responses.

---

> > > > > ### Author Response · Authors · 2024-11-30
> > > > > **Thanks for acknowledging our responses**
> > > > >
> > > > > Dear Reviewer Svzf,
> > > > >
> > > > > Thank you for your time, effort, and positive evaluation. Your guidance and insights are invaluable to our research. We remain committed to addressing any outstanding concerns and ensuring the improvement of the quality of our work.
> > > > >
> > > > > We once again appreciate your thoughtful feedback and guidance.
> > > > >
> > > > > Sincerely,
> > > > >
> > > > > All Authors

---

### Official Review · Reviewer_UFQG · 2024-11-03

**Soundness:** 2
**Presentation:** 3
**Contribution:** 2
**Rating:** 5
**Confidence:** 5

**Summary:**

The paper presents Tree Preference Optimization (TPO), an enhancement to Direct Preference Optimization (DPO) for LLM alignment. Unlike DPO’s binary preference structure, TPO uses ranked preference lists and an Adaptive Step Reward to adjust the reward margin, allowing the model to learn from multi-branch and multi-step reasoning paths and improve accuracy in complex tasks.

**Strengths:**

1.	The paper is well-written and easy to follow.
2.	The paper proposes using the classical Lambda weight from Learning-to-Rank (LTR) to construct the TPO loss, which enhances the accuracy of list ranking. This is an interesting design.
3.	The design of a loss function for multi-branch and multi-step scenarios is a highly significant research direction.The paper makes several attempts in this direction.

**Weaknesses:**

1. The most critical issue is that I believe obtaining noise-free, list-wise preference data is very costly. A more essential challenge is how to reliably construct such data. The construction method in the paper likely introduces substantial noise, and I am skeptical about the positive impact of aligning the llm to a noisy list ranking.
2. The Adaptive Step Reward mechanism proposed is overly intuitive, lacking more rigorous experimental analysis and theoretical support.
3. The experimental design in the paper is somewhat unfair, as DPO only uses pair-wise data, whereas TPO leverages additional augmented data. I believe the list data should also be converted into pair-wise form, allowing for a fair comparison. This would enable analysis of whether list-wise data is indeed more effective than pair-wise when the total information content is kept constant.

**Questions:**

See Weakness

---

> ### Author Response · Authors · 2024-11-22
> **Response to Reviewer UFQG (1/3)**
>
> Thanks a lot for your acknowledgment of the writing, presentation and novelty of our submission!
>
> **Response to W1** about the data construction method. We acknowledge that the data construction method provided in the manuscript has certain limitations, which are **discussed in detail in Line 517 of the original manuscript**. Nevertheless, we still believe that **our data construction method is reliable**, for the following reasons:
>
>
> - In the experimental setup of this work, our goal is to align open-source models with GPT-4, the current strongest long-chain reasoning models. The performance of GPT-4 in long-chain reasoning has been validated by our experiments (Tables 1 and 2 of manuscript). In this alignment process, we propose a more effective method, TPO.
>
> - Existing works [1,2,3] have shown that using GPT to construct data and iteratively enhance open-source models is a **common practice** in the academic community, which validates the reliability of our approach.

---

> ### Author Response · Authors · 2024-11-22
> **Response to Reviewer UFQG (2/3)**
>
> **Response to W2** about the discussion of *Adaptive Step Reward*. Thank you for your suggestions. We are pleased to discuss our further experiments and insights on the *Adaptive Step Reward* with you.
>
> - *T-test on Adaptive Step Reward*. To assess the statistical significance of our proposed *Adaptive Step Reward*, we performed 20 independent experiments for both TPO and TPO w/o *Adaptive Step Reward* under identical experimental conditions. The performance metrics from these experiments were then analyzed using an independent t-test to compare the means and the standard error of the two groups. The experimental results are shown in the table below, where $\^{*}$ denotes $p<0.01$, and $\^{+}$ denotes $p<0.05$.
>
>
> | Methods                  | MATH       | SVAMP      | ASDiv      | GSM-Plus   |
> |--------------------------|------------|------------|------------|------------|
> | Qwen2-7B-Instruct        |            |            |            |            |
> | TPO                      | 55.54±0.06$\^{*}$ | 48.00±0.06$\^{+}$ | 59.25±0.11$\^{*}$ | 54.86±0.04$\^{*}$ |
> | w/o Adaptive Step Reward | 55.18±0.07 | 47.80±0.07 | 58.60±0.15 | 54.33±0.05 |
> | p-value                  | 0.0004     | 0.0361     | 0.0013     | 1.0970e-9  |
> | DeepSeekMath-7B-Instruct |            |            |            |            |
> | TPO                      | 51.31±0.09$\^{*}$ | 86.85±0.07$\^{+}$ | 90.83±0.11$\^{+}$ | 64.67±0.04$\^{*}$ |
> | w/o Adaptive Step Reward | 50.78±0.08 | 86.55±0.10 | 90.46±0.08 | 64.34±0.04 |
> | p-value                  | 9.9973e-5  | 0.0167     | 0.0129     | 2.8655e-6  |
>
> The null hypothesis assumed no significant difference in performance between TPO and TPO w/o *Adaptive Step Reward*. A p-value less than 0.01/0.05 was considered statistically significant, indicating that the observed differences were unlikely to occur by chance. The results showed that **TPO consistently outperformed the TPO w/o *Adaptive Step Reward***, with statistically significant improvements in pass@1 accuracy, validating the effectiveness of *Adaptive Step Reward*.
>
>
>
> - *Case study on Adaptive Step Reward*. We further clarify our inspiration for *Adaptive Step Reward* through a case study. The following case is provided:
>
> ```
> **Prompt:**
> Let $A=2+i$, $B=3-2i$, $O=-4$, $P=-i$, $Q=5+3i$, $S=5-3i$, and $T=-3-6i$. Calculate $(A+B-O+P+Q*S)+T$.
> Let's think step by step.
> Step 1:
> First, we need to find the sum of all the complex numbers in the problem.
> We are given $A=2+i$, $B=3-2i$, $O=-4$, $P=-i$, $Q=5+3i$, $S=5-3i$, and $T=-3-6i$.
> We need to find $(A+B-O+P+Q*S)+T$.
> Step 2:
>
> **Response_1**
> Let's add all the complex numbers together.
> First, add $A$ and $B$:
> $A+B = (2+i) + (3-2i) = 2+3+i-2i = 5-i$
> Then, subtract $O$:
> $5-i - (-4) = 5-i+4 = 9-i$
> Next, add $P$:
> $9-i + (-i) = 9-2i$
> Then, calculate $Q*S$:
> $(5+3i) * (5-3i) = 25-15i+15i-9 = 16$
> Next, add $Q*S$:
> $9-2i + 16 = 25-2i$
> Finally, add $T$:
> $25-2i + (-3-6i) = 25-2i-3-6i = 22-8i$
> So, $(A+B-O+P+Q+S)+T = 22-8i$.
>
> **Response_2**
> First, let's add all the given complex numbers:
> $A+B = (2+i) + (3-2i) = 5-i$,
> $-O = -(-4) = 4$,
> $P = -i$,
> $Q*S = (5+3i) * (5-3i) = 25+15i-15i-9 = 16$,
> $T = -3-6i$.
> Now, let's add these results together:
> $5 - i + 4 - i + 16 -3 -6i = 22 - 8i$.
>
> **Response_3**
> Let's add all the complex numbers together.
> First, add $A$ and $B$:
> $A+B = (2+i) + (3-2i) = 2+3+i-2i = 5-i$
> Then, subtract $O$:
> $5-i - (-4) = 5-i+4 = 9-i$
> Next, add $P$:
> $9-i + (-i) = 9-2i$
> Then, add $Q$:
> $9-2i + (5+3i) = 9-2i+5+3i = 14+i$
> Next, multiply $S$:
> $(14+i) * (5-3i) = 70+5i-42i-3 = 67-37i$
> Finally, add $T$:
> $67-37i + (-3-6i) = 67-37i-3-6i = 64-43i$
> So, $(A+B-O+P+Q*S)+T = 64-43i$.
> ```
>
> It is noteworthy that $Responses_1$ and $Responses_2$, despite having significantly different expressions, both follow the correct steps, whereas $Response_3$, though similar to $Response_1$ in expression, fails to prioritize multiplication and thus yields an incorrect result. In the ground truth ranking, the preference ranking is $Response_1 \succ Response_2 \succ Response_3$. However, during LLMs’ preference learning, since implicit reward are computed token by token, the significant overlap of tokens between $Response_3$ and $Response_1$ leads to an incorrect ranking of $Response_1 \succ Response_3 \succ Response_2$.
>
> Based on this observation, we propose the *Adaptive Step Reward* mechanism leveraging step-level semantic similarity to adjust the reward margin between preferences. **We emphasize the importance of semantics in preference ranking and believe this is particularly critical in tasks like mathematical reasoning, which emphasize semantic similarity rather than token overlap.**
>
> We sincerely invite you to review the analysis provided in **Line 1026 of the revised manuscript**, where we present additional case studies and demonstrate the success of the *Adaptive Step Reward* in these cases. These examples help to further elucidate the insights and effectiveness of our *Adaptive Step Reward* approach.

---

> ### Author Response · Authors · 2024-11-22
> **Response to Reviewer UFQG (3/3)**
>
> **Response to W3** about the fairness of the comparison between DPO and TPO.
>
> Thank you very much for your valuable suggestions. We acknowledge that this was an oversight in our experimental setup, and we believe this is a very crucial experiment.
>
> We introduce two new baselines, which extract paired data from the preference list and applies the DPO algorithm. Specifically, given data $[x, y_1, y_2, ..., y_n]$ with the preference ranking $y_1 \succ y_2 \succ, ... , \succ y_n$, we extract two types of data: $[x, (y_1, y_2), (y_1, y_3), ..., (y_1, y_n)]$ (denoted as $DPO^{*}$) and $[x, (y_1, y_2), (y_2, y_3), ..., (y_{n-1}, y_n)]$ (denoted as $DPO^{+}$). We conducted experiments on `Qwen2-7B-Instruct` and `DeepSeekMath-7B-Instruct`, and the experimental results are shown in the table below. The experimental results indicate that **TPO outperforms all the baseline algorithm**.
>
>
> | Methods                        | MATH  | SVAMP | ASDiv | GSM-Plus | Avg.  |
> |--------------------------------|-------|-------|-------|----------|-------|
> | Qwen2-7B-Instruct+DPO          | 54.26 | 44.69 | 54.32 | 50.28    | 50.89 |
> | Qwen2-7B-Instruct+$DPO^{*}$        | 55.02 | 46.85 | 57.96 | 53.26    | 53.27 |
> | Qwen2-7B-Instruct+$DPO^{+}$        | 54.78 | 46.13 | 55.30 | 50.31    | 51.63 |
> | Qwen2-7B-Instruct+TPO          | **55.46** | **48.20** | **59.22** | **54.82**    | **54.43** |
> | DeepSeekMath-7B-Instruct+DPO   | 48.66 | 85.98 | 90.16 | 62.86    | 71.92 |
> | DeepSeekMath-7B-Instruct+$DPO^{*}$ | 50.33 | 86.20 | **90.61** | 63.57    | 72.68 |
> | DeepSeekMath-7B-Instruct+$DPO^{+}$ | 49.75 | 85.78 | 89.78 | 63.11    | 72.11 |
> | DeepSeekMath-7B-Instruct+TPO   | **51.30** | **86.80** | **90.61** | **64.73**    | **73.36** |
>
> Our **perspective** is as follows:
> - We argue that all variants of DPO focus solely on the relative likelihood between chosen and rejected preferences. However, this approach causes the likelihood of the chosen preferences to decrease during the optimization process [4,5]. In contrast, TPO introduces lambda weights into the ranking algorithm to provide absolute positional information for preferences within the list, mitigating data likelihood decline issues.
>
> [1] "Visual Instruction Tuning." NeurIPS (2023).
>
> [2] "GPT Assisted Annotation of Rhetorical and Linguistic Features for Interpretable Propaganda Technique Detection in News Text." WWW (2024).
>
> [3] "Emovit: Revolutionizing emotion insights with visual instruction tuning." CVPR (2024).
>
> [4] "Noise contrastive alignment of language models with explicit rewards." arXiv preprint arXiv:2402.05369 (2024).
>
> [5] "Smaug: Fixing failure modes of preference optimisation with dpo-positive." arXiv preprint arXiv:2402.13228 (2024).
>
>
>
> We have added the additional case study analysis and the extra baseline algorithms ($DPO^{*}$ and $DPO^{+}$) to the revised manuscript. To assist you in quickly locating our modifications, we have highlighted the changes using brown and stone blue fonts. (Please refer to the *Guideline for Reviewers* on the last page of the revised manuscript.)
>
> We want to express our sincere gratitude for your review. If you have any further questions or concerns, please feel free to contact us at any time. We are always available and look forward to further discussions with you.
>
> Best regards,
>
> All Authors

---

> ### Author Response · Authors · 2024-11-25
> **Kindly Request for Reviewer's Feedback**
>
> Dear Reviewer UFQG,
>
> We understand that you may be busy, and we sincerely appreciate your efforts in reviewing our paper. We have made significant efforts to address the concerns you raised during the initial review, but we have not yet received any feedback.
>
> As the rebuttal phase is nearing its end, we would like to kindly follow up. We would greatly appreciate it if you could share your thoughts on our responses. If there are still any issues or areas that require further improvement, we are more than willing to continue refining our work.
>
> Thank you again for your time and valuable feedback!

---

### Official Review · Reviewer_2RYw · 2024-11-03

**Soundness:** 3
**Presentation:** 3
**Contribution:** 2
**Rating:** 8
**Confidence:** 4

**Summary:**

This work proposed TPO which enables direct preference alignment for long-chain reasoning tasks.  By sampling with tree-of-thoughts, lists of generated CoTs would be paired with preference signals, which are used for LLM alignment. Limited by the original DPO, which only enables pairwise preference optimization, TPO proposed list ranking formulation to extend DPO framework. Empirical studies on several mathematical reasoning tasks show better performance.

**Strengths:**

1. This paper is well motivated and well presented.

2. The connection between LLM alignment and learn-to-rank problems is well explained.

3. The proposed implicit reward model formulation in Eq(10) is novel.

**Weaknesses:**

1. Regarding the preference list ranking, there might have been some studies on its direct optimization approaches [1-3]. The authors could consider discussing their unique challenges or difference to those works. Ideally, the authors might consider adapting their ideas as baselines for a more comprehensive comparison.

2. The backbone general LLMs are restricted to only one type, Qwen. The authors might considering experimenting on more types of LLMs, including Llama-3, Phi-2, Mistral, etc.

3. The authors could consider including supervised fine-tuning and/or instruction fine-tuning baselines, which could help the readers better understand the potential performance upper bound on the specific downstream tasks.

4. It would be critical for the authors to also demonstrate the generalizability of the LLMs after TPO on general benchmark or metrics, which might prevent potential chances of model overfitting on the downstream tasks.

5. Although the proposed DPO is proposed for general chain-of-thought reasoning, only maths and coding tasks are evaluated. More commensense, multi-hop question-answering, and especially knowledge-intensive tasks could be evaluated.

[1] Chen, Yuxin, et al. "On Softmax Direct Preference Optimization for Recommendation." arXiv preprint arXiv:2406.09215 (2024).

[2] Liao, Jiayi, et al. "RosePO: Aligning LLM-based Recommenders with Human Values." arXiv preprint arXiv:2410.12519 (2024).

[3] Scheid, Antoine, et al. "Optimal Design for Reward Modeling in RLHF." arXiv preprint arXiv:2410.17055 (2024).

**Questions:**

1. Could the authors explain the link between DPO on chain-of-thought reasoning, which from my understanding is the major contribution of this paper, and the proposed Preference List Ranking optimization. Specifically, I was wondering if extracting pairwise samples from the ranking list could simply solve the challenge?

2. Could the authors explain their means of acquisition of such ranking list preference signals, which could be relatively more expensive and harder to find? Any intensive human annotation process involved in this paper?

3. Could the authors try to discuss the potential alternative ranking list preference alignment methods I provided in Weaknesses?

4. Could the authors provide some insights or experimental results, if applying their TPO on knowledge-intensive CoTs? Since in such tasks knowledge fidelity (factuality) could be entangled with logical reasoning steps (tree-of-thought sampling), the alignment process is not as straightforward as mathematical or coding problems. In such cases, would TPO still work under some guarantees with some lower-bounded confidence?

---

> ### Author Response · Authors · 2024-11-22
> **Response to Reviewer 2RYw (1/4)**
>
> Thanks a lot for your acknowledgment of the motivation, presentation and novelty of our submission!
>
> **Response to W1 and Q3** about the related literature on preference list ranking.
> - We present the optimization objectives of TPO and related literature [1,2,3] you provided as follows:
>
> S-DPO [1]:
> $$
> -\log\sigma\left(-\log\sum_{y_{l}\in\mathcal{Y}_{l}}\exp\left(\beta\log\frac{\pi _ {\theta}(y _ {w}|x)}{\pi _ {\mathrm{ref}}(y _ {w}|x)}-\beta\log\frac{\pi _ {\theta}(y _ {l}|x)}{\pi _ {\mathrm{ref}}(y _ {l}|x)}\right)\right)
> $$
>
> RoseDPO [2]:
> $$
> -(1-\epsilon_{\phi})\left(\beta\log\frac{\pi_{\theta}(y_{w}|x)}{\pi_{\mathrm{ref}}(y_{\mathrm{w}}|x)}-\beta\log\frac{\pi_{\theta}(y_{l}|x)}{\pi_{\mathrm{ref}}(y_{l}|x)}\right) -\epsilon_{\phi}\left(\beta\log\frac{\pi_{\theta}(y_{l}|x)}{\pi_{\mathrm{ref}}(y_{l}|x)}-\beta\log\frac{\pi_{\theta}(y_{\mathrm{w}}|x)}{\pi_{\mathrm{ref}}(y_{\mathrm{w}}|x)}\right)
> $$
> Where $\epsilon_{\phi}$ represents the uncertainty assessment of preferences.
>
> ODPO [3]:
> $$
> -\log\prod_{i=1}^{N}\frac{\exp(\beta\operatorname{log}\frac{\pi_{\theta}(y_{i}|x)}{\pi_{\mathrm{ref}}(y_{i}|x)})}{\sum_{j=i}^{N}\exp(\beta\operatorname{log}\frac{\pi_{\theta}(y_{j}|x)}{\pi_{\mathrm{ref}}(y_{j}|x)})}
> $$
>
> TPO (Ours):
> $$
> -\lambda_{i,j}\sum_{v_{i}>v_{j}}\log\sigma(\beta\operatorname{log}\frac{\pi_{\theta}(y_{i}|x)}{\pi_{\mathrm{ref}}(y_{i}|x)}-\beta\operatorname{log}\frac{\pi_{\theta}(y_{j}|x)}{\pi_{\mathrm{ref}}(y_{j}|x)})
> $$
>
> **TPO vs. S-DPO**: S-DPO [1] employs SoftMax to maximize the reward margin between chosen preference and all rejected preferences. However, S-DPO does not account for the contrasts between rejected preferences, nor does it consider the absolute positional information of preferences within the list.
>
> **TPO vs. Rose-DPO**: On the one hand, Rose-DPO [2] does not consider the ranking of the preference list but instead incorporates the uncertainty of preferences. On the other hand, similar to TPO, both Rose-DPO and TPO use additional weights to adjust the reward margin. However, while Rose-DPO is based on uncertainty, TPO adjusts the margin based on the semantic similarity between steps.
>
>
> **TPO vs. ODPO**: ODPO [3] utilizes Maximum Likelihood Estimation (MLE) to optimize list-wise preferences. However, existing Learn-to-Rank literature [4,5] suggests that list-MLE is not an ideal ranking optimization objective, as it enforces strict list ordering without considering the actual label values.
>
> - It is noteworthy that references [2,3] were posted on arXiv on October 16, 2024, and October 22, 2024, respectively. The submission deadline for ICLR 2025 was October 1, 2024.

---

> ### Author Response · Authors · 2024-11-22
> **Response to Reviewer 2RYw (2/4)**
>
> **Response to W2 and W3** about more comprehensive comparison. We provide more extensive experimental results, as shown in the table below.
>
> | LLMs                         | size | MATH  | SVAMP | ASDiv | GSM-Plus | Avg.  |
> |------------------------------|------|-------|-------|-------|----------|-------|
> | Qwen2-1.5B-Instruct          | 1.5B | 19.52 | 23.90 | 35.76 | 20.05    | 24.81 |
> | Qwen2-1.5B-Instruct+SFT      | 1.5B | 20.80 | 28.77 | 38.32 | 21.87    | 27.44 |
> | Qwen2-1.5B-Instruct+DPO      | 1.5B | 20.98 | 29.30 | 40.13 | 21.52    | 27.98 |
> | Qwen2-1.5B-Instruct+TPO      | 1.5B | **22.88** | **35.60** | **46.28** | **24.12**    | **32.22** |
> | Qwen2-7B-Instruct            | 7B   | 53.92 | 33.90 | 48.38 | 44.72    | 45.23 |
> | Qwen2-7B-Instruct+SFT        | 7B   | 54.92 | 46.40 | 53.28 | 45.40    | 50.00 |
> | Qwen2-7B-Instruct+DPO        | 7B   | 54.26 | 44.69 | 54.32 | 50.28    | 50.89 |
> | Qwen2-7B-Instruct+TPO        | 7B   | **55.46** | **48.20** | **59.22** | **54.82**    | **54.43** |
> | DeepSeekMath-7B-Instruct     | 7B   | 43.92 | 84.10 | 90.10 | 59.78    | 69.48 |
> | DeepSeekMath-7B-Instruct+SFT | 7B   | 45.18 | 84.30 | 90.45 | 61.09    | 70.26 |
> | DeepSeekMath-7B-RL           | 7B   | 50.70 | 86.70 | **90.94** | 64.07    | 73.10 |
> | DeepSeekMath-7B-Instruct+DPO | 7B   | 48.66 | 85.98 | 90.16 | 62.86    | 71.92 |
> | DeepSeekMath-7B-Instruct+TPO | 7B   | **51.30** | **86.80** | 90.61 | **64.73**    | **73.36** |
> | Meta-Llama-3-8B-Instruct     | 8B   | 28.50 | 74.30 | 75.57 | 48.77    | 56.79 |
> | Meta-Llama-3-8B-Instruct+SFT | 8B   | 28.08 | 73.30 | 74.92 | 47.50    | 55.95 |
> | Meta-Llama-3-8B-Instruct+DPO | 8B   | 29.12 | 73.10 | 75.73 | 47.66    | 56.40 |
> | Meta-Llama-3-8B-Instruct+TPO | 8B   | **29.58** | **77.70** | **80.91** | **53.25**    | **60.36** |
> | Mistral-7B-Instruct-v0.3     | 7B   | 14.30 | 53.60 | 60.36 | 30.97    | 39.81 |
> | Mistral-7B-Instruct-v0.3+SFT | 7B   | 14.62 | 56.00 | 62.46 | 33.60    | 41.67 |
> | Mistral-7B-Instruct-v0.3+DPO | 7B   | 15.98 | 70.44 | 71.85 | 38.62    | 49.22 |
> | Mistral-7B-Instruct-v0.3+TPO | 7B   | **17.56** | **72.90** | **75.24** | **41.59**    | **51.82** |
>
>
> - *More general LLMs.* Following your suggestion, we further introduced `Meta-Llama-3-8B-Instruct` and `Mistral-7B-Instruct-v0.3` as general-purpose LLM backbones. Experimental results demonstrate that **TPO consistently outperforms the DPO algorithm, similar to the results observed on `Qwen2-7B-Instruct`**.
>
> - *Baseline of supervised fine-tuning (SFT).* We introduced SFT as a new baseline algorithm. The experimental results indicate that the performance of the **SFT is, in most cases, even inferior to the DPO algorithm**. This is because SFT only provides positive feedback to LLMs without suppressing incorrect reasoning paths. This limitation has been demonstrated in related work and is also illustrated in Line 40 of our original manuscript.

---

> ### Author Response · Authors · 2024-11-22
> **Response to Reviewer 2RYw (3/4)**
>
> **Response to W4, W5 and Q4** about results on general benchmark. We introduced two new evaluation datasets, BBH [6] and MMLU [7], to assess the model's performance on commonsense knowledge. Notably, BBH requires multi-step reasoning, whereas MMLU does not. The experimental results are shown in the table below.
>
>
> | LLMs                         | size | BBH   | MMLU  |
> |------------------------------|------|-------|-------|
> | Qwen2-1.5B-Instruct          | 1.5B | 32.76 | 55.97 |
> | Qwen2-1.5B-Instruct+SFT      | 1.5B | 33.10 | 55.90 |
> | Qwen2-1.5B-Instruct+DPO      | 1.5B | 34.72 | **56.03** |
> | Qwen2-1.5B-Instruct+TPO      | 1.5B | **37.49** | 55.99 |
> | Qwen2-7B-Instruct            | 7B   | 62.43 | **70.78** |
> | Qwen2-7B-Instruct+SFT        | 7B   | 64.75 | 70.74 |
> | Qwen2-7B-Instruct+DPO        | 7B   | 65.58 | 70.58 |
> | Qwen2-7B-Instruct+TPO        | 7B   | **69.62** | 70.63 |
> | DeepSeekMath-7B-Instruct     | 7B   | 61.85 | 54.44 |
> | DeepSeekMath-7B-Instruct+SFT | 7B   | 62.79 | 54.56 |
> | DeepSeekMath-7B-RL           | 7B   | 62.74 | 54.98 |
> | DeepSeekMath-7B-Instruct+DPO | 7B   | 62.68 | 54.22 |
> | DeepSeekMath-7B-Instruct+TPO | 7B   | **62.99** | **54.99** |
>
>
>
> - The experimental results indicate that **TPO achieves optimal performance on the BBH task**, which requires multi-step reasoning, outperforming all baseline algorithms. However, on the MMLU dataset, which does not require multi-step reasoning, the performance of **all algorithms is nearly identical**.
>
> - Our **perspective** is as follows: during downstream fine-tuning (SFT, DPO, TPO), LLMs acquire two types of knowledge, including mathematical deduction and long-chain reasoning. The long-chain reasoning knowledge enables the LLMs to generalize its capabilities beyond the mathematical domain, leading to performance improvements in Coding and Reasoning tasks. For the MMLU dataset, however, since it does not require long-chain reasoning and TPO fine-tuning does not introduce new commonsense knowledge, the performance of LLMs fine-tuned with TPO does not improve further. Notably, **TPO does not lead to performance degradation of LLMs on MMLU tasks, indicating that commonsense knowledge is preserved during the TPO fine-tuning process.**
>
>
> **Response to Q1** about the link between DPO/TPO and CoT reasoning.
>
> - The motivation for applying DPO to chain-of-thought reasoning is that we aim to fine-tune LLMs using downstream task data. However, conventional supervised fine-tuning (SFT) only provides positive feedback and does not suppress erroneous outputs from LLMs, which is particularly critical in long-chain reasoning. An error at any step can lead to incorrect results for all subsequent steps. In contrast to DPO, TPO provides preference list ranking optimization, which can accommodate a broader range of preferences. By leveraging the preference relationships provided in the ranking, TPO aligns LLMs more effectively.
>
> - We introduce two new baselines, which extract paired data from the preference list and applies the DPO algorithm. Specifically, given data $[x, y_1, y_2, ..., y_n]$ with the preference ranking $y_1 \succ y_2 \succ, ... , \succ y_n$, we extract two types of data: $[x, (y_1, y_2), (y_1, y_3), ..., (y_1, y_n)]$ (denoted as $DPO^{*}$) and $[x, (y_1, y_2), (y_2, y_3), ..., (y_{n-1}, y_n)]$ (denoted as $DPO^{+}$). We conducted experiments on `Qwen2-7B-Instruct` and `DeepSeekMath-7B-Instruct`, and the experimental results are shown in the table below. The experimental results indicate that **TPO outperforms all the baseline algorithm**.
>
>
> | Methods                        | MATH  | SVAMP | ASDiv | GSM-Plus | Avg.  |
> |--------------------------------|-------|-------|-------|----------|-------|
> | Qwen2-7B-Instruct+DPO          | 54.26 | 44.69 | 54.32 | 50.28    | 50.89 |
> | Qwen2-7B-Instruct+$DPO^{*}$        | 55.02 | 46.85 | 57.96 | 53.26    | 53.27 |
> | Qwen2-7B-Instruct+$DPO^{+}$        | 54.78 | 46.13 | 55.30 | 50.31    | 51.63 |
> | Qwen2-7B-Instruct+TPO          | **55.46** | **48.20** | **59.22** | **54.82**    | **54.43** |
> | DeepSeekMath-7B-Instruct+DPO   | 48.66 | 85.98 | 90.16 | 62.86    | 71.92 |
> | DeepSeekMath-7B-Instruct+$DPO^{*}$ | 50.33 | 86.20 | **90.61** | 63.57    | 72.68 |
> | DeepSeekMath-7B-Instruct+$DPO^{+}$ | 49.75 | 85.78 | 89.78 | 63.11    | 72.11 |
> | DeepSeekMath-7B-Instruct+TPO   | **51.30** | **86.80** | **90.61** | **64.73**    | **73.36** |
>
> Our **perspective** is as follows:
> - We argue that all variants of DPO focus solely on the relative likelihood between chosen and rejected preferences. However, this approach causes the likelihood of the chosen preferences to decrease during the optimization process [8,9]. In contrast, TPO introduces lambda weights into the ranking algorithm to provide absolute positional information for preferences within the list, mitigating data likelihood decline issues.

---

> ### Author Response · Authors · 2024-11-22
> **Response to Reviewer 2RYw (4/4)**
>
> **Response to Q2** about the data construction method.
>
> - We provide the detailed construction method for the training data in **Line 311 of the original manuscript**. Specifically, we use pure GPT-4 to construct the preference rankings without any manual annotations. It is worth noting that the performance of GPT-4, as verified by our experiments (Table 1 and Table 2 of manuscript), is the strongest for long-chain reasoning tasks. In addition, existing works [10,11,12] have shown that using GPT to construct data and iteratively enhance open-source models is a **common practice** in the academic community, which validates the reliability of our approach.
>
> [1] "On Softmax Direct Preference Optimization for Recommendation." arXiv preprint arXiv:2406.09215 (2024).
>
> [2] "RosePO: Aligning LLM-based Recommenders with Human Values." arXiv preprint arXiv:2410.12519 (2024).
>
> [3] "Optimal Design for Reward Modeling in RLHF." arXiv preprint arXiv:2410.17055 (2024).
>
> [4] "The lambdaloss framework for ranking metric optimization." CIKM (2018).
>
> [5] "On optimizing top-k metrics for neural ranking models." SIGIR (2022).
>
> [6] "Challenging big-bench tasks and whether chain-of-thought can solve them." ACL (2023).
>
> [7] "Measuring coding challenge competence with apps." NeurIPS (2021).
>
> [8] "Noise contrastive alignment of language models with explicit rewards." arXiv preprint arXiv:2402.05369 (2024).
>
> [9] "Smaug: Fixing failure modes of preference optimisation with dpo-positive." arXiv preprint arXiv:2402.13228 (2024).
>
> [10] "Visual Instruction Tuning." NeurIPS (2023).
>
> [11] "GPT Assisted Annotation of Rhetorical and Linguistic Features for Interpretable Propaganda Technique Detection in News Text." WWW (2024).
>
> [12] "Emovit: Revolutionizing emotion insights with visual instruction tuning." CVPR (2024).
>
>
> We have incorporated the discussion of related works (S-DPO, Rose-DPO and ODPO), additional baseline algorithms (SFT, $DPO^{*}$ and $DPO^{+}$), more general backbone LLMs (`Meta-Llama-3-8B-Instruct` and `Mistral-7B-Instruct-v0.3`), and additional benchmarks (BBH and MMLU) into the revised manuscript. To facilitate quick identification of our modifications, we have highlighted the changes using orange and brown fonts. (Please refer to the *Guideline for Reviewers* on the last page of the revised manuscript.)
>
> We want to express our sincere gratitude for your review. If you have any further questions or concerns, please feel free to contact us at any time. We are always available and look forward to further discussions with you.
>
> Best regards,
>
> All Authors

---

> ### Author Response · Authors · 2024-11-25
> **Kindly Request for Reviewer's Feedback**
>
> Dear Reviewer 2RYw,
>
> We understand that you may be busy, and we sincerely appreciate your efforts in reviewing our paper. We have made significant efforts to address the concerns you raised during the initial review, but we have not yet received any feedback.
>
> As the rebuttal phase is nearing its end, we would like to kindly follow up. We would greatly appreciate it if you could share your thoughts on our responses. If there are still any issues or areas that require further improvement, we are more than willing to continue refining our work.
>
> Thank you again for your time and valuable feedback!

---

> > ### Comment · Reviewer_2RYw · 2024-11-25
> >
> > Thanks for the reviewer's responses. I like this work and would love to see it published. I raised my score to 8.

---

> > > ### Author Response · Authors · 2024-11-25
> > > **Thanks for acknowledging our responses**
> > >
> > > Dear Reviewer 2RYw,
> > >
> > > Thank you so much for taking the time to review our work and for acknowledging our response and efforts. We truly appreciate your thoughtful feedback and guidance.
> > >
> > > Sincerely,
> > >
> > > All Authors

---

### Author Response · Authors · 2024-11-25
**General Rebuttal**

We would like to express our sincere gratitude to all the reviewers for their recognition of our work, particularly for the **unanimous acknowledgment** of the **presentation** and **novelty** of our manuscript. We further appreciate the insightful comments and constructive suggestions provided by the reviewers, which have played a **crucial role** in enhancing the quality of our manuscript.

In summary, we have followed the reviewers' recommendations by providing additional experiments and discussions to clarify the effectiveness and correctness of TPO. The details are as follows:

1. **Verification of TPO’s performance across more general LLMs, benchmarks, and additional baseline comparisons** (Reviewer 2RYw for W2, W3, W4, W5 and Q4):
    - TPO consistently outperforms existing baseline algorithms and demonstrates robustness across multiple general-purpose LLMs.
    - See `Tables 1, 2, and 4` in the revised manuscript. (Line 379, 401 and 918)

2. **Comparison with enhanced DPO** (Reviewer 2RYw for Q1, Reviewer UFQG for W3):
    - TPO surpasses baseline models and shows that regardless of the enhancement in DPO, the method has inherent limitations when only optimizing relative likelihoods.
    - See `Tables 5` in the revised manuscript. (Line 934)

3. **Effectiveness of *Adaptive Step Rewards*** (Reviewer UFQG for W2, Reviewer Svzf for W2 and W3):
    - We present t-test experiments to statistically analyze the effectiveness of *Adaptive Step Reward*.
    - Through case studies, we explain the motivations behind our approach to *Adaptive Step Reward* and validate their effectiveness by observing reward margins.
    - We emphasize the significance of semantics, especially in tasks such as mathematical reasoning.
    - See `C.4` (Line 1013) and `C.5` (Line 1026) in the revised manuscript.

5. **Data construction methods** (Reviewer 2RYw for Q2, Reviewer UFQG for W1):
    - Through a review of existing literature, we summarize that using GPT-4 for data construction and enhancing open-source models is a **common practice** in the academic community.

6. **Further exploration of ranking loss** (Reviewer Svzf for W1):
    - Experimental results validate the correctness and effectiveness of LambdaLoss.
    - See `Table 6` in the revised manuscript. (Line 927)

7. **Discussion with Related literature** (Reviewer 2RYw for W1 and Q3):
    - We highlight the differences in the optimization objectives between TPO and the related literature, emphasizing the advantages of TPO in terms of preference absolute positioning.
    - See `Table 8` in the revised manuscript. (Line 1156)

For the additional experiments mentioned above, we provide further analysis and insights in our responses to the reviewers. We confirm that **the new experimental results continue to support the primary conclusions of the original manuscript**. All experiments have been incorporated into the revised manuscript, with different colored fonts used to distinguish responses to different reviewers.

Once again, we sincerely thank all the reviewers for their time and effort in reviewing our manuscript. Should there be any further questions or comments, we are always available and eager to engage in further discussions.

Best regards,

All Authors

---

### Meta-Review · Area_Chair_2AEU · 2024-12-20

**Metareview:**

This paper introduces TPO, which learns directly from the entire preference tree during fine-tuning rather than sampling paired preference responses. TPO formulates language model alignment as a preference list ranking problem and employs adaptive step reward to enhance the reward values at each step, enabling fine-grained preference optimization. The paper is well-written, and the authors address almost all of the reviewers' concerns.

**Additional Comments On Reviewer Discussion:**

Reviewer UFQG is not responsive on the authors rebuttal text. After checking the authors' response, I think they fixed the questions.

---

### Decision · Program_Chairs · 2025-01-22

Accept (Poster)